# ON A CONNECTION BETWEEN IMITATION LEARNING AND RLHF

**Teng Xiao♠, Yige Yuan♣, Mingxiao Li♠, Zhengyu Chen♦, Vasant G Honavar♠**
♠Pennsylvania State University ♣University of Chinese Academy of Sciences
♠Tencent AI Lab ♦Meituan Inc
tengxiao@psu.edu, vhonavar@psu.edu

## ABSTRACT

This work studies the alignment of large language models with preference data from an imitation learning perspective. We establish a close theoretical connection between reinforcement learning from human feedback (RLHF) and imitation learning (IL), revealing that RLHF implicitly performs imitation learning on the preference data distribution. Building on this connection, we propose DIL, a principled framework that directly optimizes the imitation learning objective. DIL provides a unified imitation learning perspective on alignment, encompassing existing alignment algorithms as special cases while naturally introducing new variants. By bridging IL and RLHF, DIL offers new insights into alignment with RLHF. Extensive experiments demonstrate that DIL outperforms existing methods on various challenging benchmarks. The code for DIL is available at https://github.com/tengxiao1/DIL.

## 1 INTRODUCTION

Aligning large language models (LLMs) with human preferences is essential to ensure that the responses generated by LLMs align with human expectations (Bai et al., 2022; Ouyang et al., 2022; Stiennon et al., 2020). Recently, Reinforcement Learning from Human Feedback (RLHF) (Ouyang et al., 2022; Christiano et al., 2017) has become a widely adopted framework for fine-tuning language models according to human preference data. This approach typically involves training a reward model based on human feedback and subsequently employing reinforcement learning (RL) algorithms, such as PPO (Schulman et al., 2017), to optimize the model to maximize the reward signal.

RLHF has demonstrated impressive efficacy across a diverse range of tasks, from programming to creative writing. However, its dependence on two-step reinforcement learning presents challenges, such as computational inefficiency and instability during training (Engstrom et al., 2020; Rafailov et al., 2024). To mitigate these limitations, alternative one-step approaches, such as direct preference optimization (DPO) and its variants, have been proposed (Rafailov et al., 2024; Meng et al., 2024; Tajwar et al., 2024). These methods replace RLHF with supervised learning, eliminating the need for explicit reward modeling. Instead, they directly define an implicit reward based on the likelihood of preference data, resulting in significant gains in efficiency while preserving competitive performance.

While DPO theoretically aims to discover identical optimal policies as RLHF, it and its variants fundamentally adhere to the reward maximization objective and are determined by parametric models such as the Bradley-Terry (BT) model (Bradley & Terry, 1952), making them prone to overfitting (Yuan et al., 2024; Xiao et al., 2024b) and resulting in suboptimal alignment with preference data (Xu et al., 2024c; Wu et al., 2024). This raises a fundamental and open research question: *Can we understand and design effective preference optimization algorithms from a new perspective?*

In this paper, we revisit RLHF from the perspective of imitation learning. In particular, we show that RLHF is a special case of a general imitation learning problem expressed exclusively in terms of pairwise preferences. We theoretically demonstrate that alignment with RLHF closely resembles imitation learning and implicitly optimizes the same objective. We then leverage this insight to design DIL, a general framework for effective alignment based on density ratio reward estimation.

Our primary technical contributions are as follows: (i) We prove that RLHF for alignment is essentially an imitation learning problem and provide a novel analysis that offers explicit guidance for alignemnt

algorithm design. (ii) We propose `DIL`, a simple and generalized imitation learning framework for alignment. `DIL` unifies imitation learning on preference data and bridges the gap between density ratio estimation and preference alignment. (iii) Empirically, we validate the effectiveness of `DIL` on widely used benchmarks, demonstrating that it outperforms previous alignment methods.

## 2 RELATED WORK

**Reinforcement Learning from Human Feedback.** `RLHF` has emerged as an effective approach for aligning LLMs with human preferences (Christiano et al., 2017). It involves first training a reward model from human feedback through supervised learning, which is then used to optimize a policy via RL algorithms, such as PPO (Schulman et al., 2017). `RLHF` has been successfully applied to a wide range of tasks, including summarization (Stiennon et al., 2020), instruction following (Ouyang et al., 2022), safety improvement (Bai et al., 2022), and truthfulness enhancement (Tian et al., 2023).

**Offline Preference Optimization.** Recent literature highlights the inherent complexity of `RLHF`, prompting the search for more efficient offline alternatives. A significant advancement in this area is `DPO` (Rafailov et al., 2024). Unlike `RLHF`, which first learns an explicit reward model and then fits the policy to rewards, `DPO` bypasses this second approximation by directly learning a policy from collected data, without the need for reward modeling. `DPO` implicitly optimizes the same objective as existing `RLHF` algorithms, but it is simpler to implement and more straightforward to train. Other offline alignment methods, such as IPO (Azar et al., 2024), KTO (Ethayarajh et al., 2024), and others (Zhao et al., 2023; Xiao & Wang, 2021; Xu et al., 2024a; Meng et al., 2024; Xiao et al., 2025), have also been proposed. In contrast, we rethink and design the alignment objective from a novel offline imitation learning perspective. We show that `RLHF` can theoretically be viewed as imitation learning, which fits the chosen response distribution by minimizing the reverse KL divergence.

**Imitation Learning**. Classical imitation learning (IL) methods often frame IL as inverse reinforcement learning (IRL) to better utilize expert demonstrations (Sammut et al., 1992; Abbeel & Ng, 2004). In the seminal work (Ho & Ermon, 2016), the authors introduce GAIL, which bypasses inner-loop reinforcement learning (RL) by establishing a connection between IL and generative adversarial networks (GANs) (Goodfellow et al., 2020). GAIL and its successor, AIRL (Fu et al., 2018), have made significant strides. However, these online methods typically require substantial environmental interactions, limiting their deployment in cost-sensitive or safety-sensitive domains. To address this issue, recent work on offline IL (Garg et al., 2021) focuses on learning a reward function from offline datasets to understand and generalize the intentions underlying expert behavior. IQ-Learn (Garg et al., 2021) simplifies AIRL's game-theoretic objective over policy and reward functions into an optimization over the soft Q-function, which implicitly represents both reward and policy. Recently, some works (Sun & van der Schaar, 2024; Wulfmeier et al., 2024; Xiao et al., 2024a) have applied imitation learning to the alignment of large language models. Different from the above works, in this paper, we aim at building a close theoretical connection between `RLHF` and imitation learning, revealing that `RLHF` implicitly performs imitation learning on the chosen data distribution.

## 3 NOTATIONS AND PRELIMINARIES

**Problem Setup.** Let the text sequence $\mathbf{x} = [x_1, x_2, \ldots]$ denote the input prompt, and $\mathbf{y}_w = [y_1, y_2, \ldots]$ and $\mathbf{y}_l$ denote two responses, typically sampled from the same reference policy $\pi_{\text{ref}}(\mathbf{y} \mid \mathbf{x})$. The response pairs are then presented to human labelers (or an oracle) who express preferences for responses given the prompt, denoted as $\mathbf{y}_w \succ \mathbf{y}_l \mid \mathbf{x}$, where $\mathbf{y}_w$ and $\mathbf{y}_l$ denote preferred and dispreferred responses, respectively. The preference distribution is typically expressed as:

$$p\left(\mathbf{y}_w \succ \mathbf{y}_l \mid x\right) = g\left(r(\mathbf{x}, \mathbf{y}_w) - r\left(\mathbf{x}, \mathbf{y}_l\right)\right). \tag{1}$$

where $g$ is the sigmoid function $\sigma(x) = \frac{1}{1+e^{-x}}$ based on the Bradley-Terry (BT) preference assumption (Bradley & Terry, 1952). Given a preference dataset $\mathcal{D}$, containing feedback $(\mathbf{x}, \mathbf{y}_w, \mathbf{y}_l)$, the goal of alignment is to learn an LLM policy $\pi(\mathbf{y} \mid \mathbf{x})$ based on the preference data.

**Reinforcement Learning from Human Feedback.** Given the estimated reward function $r(\mathbf{x}, \mathbf{y})$, dictating the human preferences, `RLHF` fine-tunes policy $\pi_{\boldsymbol{\theta}}$ by optimizing the following objective:

$$\max_{\pi_{\boldsymbol{\theta}}} \mathbb{E}_{\mathbf{y} \sim \pi_{\boldsymbol{\theta}}(\mathbf{y}|\mathbf{x})} \left[r(\mathbf{x}, \mathbf{y})\right] - \beta \mathbb{D}_{\text{KL}}\left[\pi_{\boldsymbol{\theta}}(\mathbf{y} \mid \mathbf{x}) \| \pi_{\text{ref}}(\mathbf{y} \mid \mathbf{x})\right], \tag{2}$$

where $\beta > 0$ is an appropriate KL penalty coefficient. RLHF typically optimizes the above objective in Equation (2) using RL algorithms, such as PPO (Ouyang et al., 2022; Schulman et al., 2017).

**Reward Modeling.** One standard approach to reward modeling is to fit a reward function $r_\phi(\mathbf{x}, \mathbf{y})$ with the BT preference model in Equation (1). Specifically, the reward function $r_\phi(\mathbf{x}, \mathbf{y})$ can be estimated by maximizing the log-likelihood over preference feedback $(\mathbf{x}, \mathbf{y}_w, \mathbf{y}_l)$:

$$\mathcal{L}_{\mathrm{RM}}(\phi; \mathcal{D}) = \mathbb{E}_{(\mathbf{x}, \mathbf{y}_w, \mathbf{y}_l) \sim \mathcal{D}} \left[ -\log \sigma \left( r_\phi(\mathbf{x}, \mathbf{y}_w) - r_\phi(\mathbf{x}, \mathbf{y}_l) \right) \right]. \tag{3}$$

**Supervised Fine-tuning (SFT).** Given a demonstration dataset, the objective of SFT is minimizing the negative log-likelihood over the demonstration data as follows:

$$\mathcal{L}_{\mathrm{SFT}}(\boldsymbol{\theta}; \mathcal{D}) = -\mathbb{E}_{(\mathbf{x}, \mathbf{y}) \sim \mathcal{D}}[\log \pi_{\boldsymbol{\theta}}(\mathbf{y} \mid \mathbf{x})]. \tag{4}$$

SFT is equivalent to behavior cloning (BC) (Pomerleau, 1988), a classical offline imitation learning method that minimizes the forward KL divergence between the learned policy and data policy:

$$\min_{\boldsymbol{\theta}} \mathrm{KL} \left( \pi_{\mathrm{data}}(\mathbf{y} \mid \mathbf{x}) \| \pi_{\boldsymbol{\theta}}(\mathbf{y} \mid \mathbf{x}) \right) = -\mathbb{E}_{\pi_{\mathrm{data}}(\mathbf{y} \mid \mathbf{x})} \left[ \log \pi_{\boldsymbol{\theta}}(\mathbf{y} \mid \mathbf{x}) \right], \tag{5}$$

It is easy to see that the BC problem above shares the same optimal solutions as SFT in expectation.

**Directed Preference Optimization.** To simplify the optimization process of RLHF, DPO uses the log-likelihood of the learning policy to implicitly represent the reward function:

$$r_{\boldsymbol{\theta}}(\mathbf{x}, \mathbf{y}) = \beta \left[ \log \pi_{\boldsymbol{\theta}}(\mathbf{y} \mid \mathbf{x}) - \log \pi_{\mathrm{ref}}(\mathbf{y} \mid \mathbf{x}) \right] + \beta \log Z_{\boldsymbol{\theta}}(\mathbf{x}), \tag{6}$$

where $Z(\mathbf{x}) = \sum_{\mathbf{y}} \pi_{\mathrm{ref}}(\mathbf{y} \mid \mathbf{x}) \exp(r_{\boldsymbol{\theta}}(\mathbf{x}, \mathbf{y})/\beta)$ is the partition function. By incorporating this reward into the BT model in Equation (1), DPO (Rafailov et al., 2024) objective enables the comparison of response pairs, facilitating the discrimination between preferred and dispreferred responses:

$$\mathcal{L}_{\mathrm{DPO}}(\boldsymbol{\theta}; \mathcal{D}) = \mathbb{E}_{(\mathbf{x}, \mathbf{y}_w, \mathbf{y}_l) \sim \mathcal{D}} \left[ -\log \sigma(\beta \log \frac{\pi_{\boldsymbol{\theta}}(\mathbf{y}_w \mid \mathbf{x})}{\pi_{\mathrm{ref}}(\mathbf{y}_w \mid \mathbf{x})} - \beta \log \frac{\pi_{\boldsymbol{\theta}}(\mathbf{y}_l \mid \mathbf{x})}{\pi_{\mathrm{ref}}(\mathbf{y}_l \mid \mathbf{x})}) \right]. \tag{7}$$

**Energy-based Models.** Energy-based models (EBMs) (LeCun et al., 2006) define the distribution through an energy function. For $\mathbf{y} \in \mathbb{R}^D$, its probability density can be expressed as follows:

$$p_{\boldsymbol{\theta}}(\mathbf{y}) = \exp(-E_{\boldsymbol{\theta}}(\mathbf{y}))/Z_{\boldsymbol{\theta}}(\mathbf{y}), \tag{8}$$

where $E_{\boldsymbol{\theta}}(\mathbf{y}) : \mathbb{R}^D \to \mathbb{R}$ is the energy function, mapping the data point $\mathbf{y}$ to a scalar, and $Z_{\boldsymbol{\theta}}(\mathbf{y}) = \sum_{\mathbf{y}} \exp(-E_{\boldsymbol{\theta}}(\mathbf{y}))$ is the unknown normalization constant (Song & Kingma, 2021).

# 4 METHODOLOGY

## 4.1 RLHF IS A FORM OF IMITATION LEARNING

In this section, we connect RLHF to the imitation learning framework. We show that RLHF is a special case of imitation learning problem on the distribution chosen response with the reverse KL divergence. Specifically, we firstly define the following policy based on EBMs (Haarnoja et al., 2017):

$$\pi_\phi(\mathbf{y} \mid \mathbf{x}) = \pi_{\mathrm{ref}}(\mathbf{y} \mid \mathbf{x}) \exp \left( r_\phi(\mathbf{x}, \mathbf{y}) \right) / Z_\phi(\mathbf{x}), \tag{9}$$

where $\phi$ denotes the parameter and $Z_\phi(\mathbf{x}) = \sum_{\mathbf{y}} \pi_{\mathrm{ref}}(\mathbf{y} \mid \mathbf{x}) \exp(r_\phi(\mathbf{x}, \mathbf{y}))$. To learn the parameter $\phi$, one can apply behavior cloning (Pomerleau, 1988), a classical and widely used imitation learning method, which frames the task as minimizing the KL divergence between the policy $\pi_\phi$ and the expert policy $\pi_{\mathrm{chosen}}$ generating the chosen response $\mathbf{y}_w$. In other works, IL learns the parameter $\phi$ such that the model distribution imitates the distribution of chosen response in the preference dataset:

$$\min_{\phi} \mathrm{KL} \left( \pi_{\mathrm{chosen}}(\mathbf{y} \mid \mathbf{x}) \| \pi_\phi(\mathbf{y} \mid \mathbf{x}) \right). \tag{10}$$

Minimizing the above forward KL divergence with the chosen responses on preference data gives us:

$$\min_{\phi} \mathbb{E}_{(\mathbf{x}, \mathbf{y}_w) \sim \mathcal{D}}[-\log \pi_{\mathrm{ref}}(\mathbf{y}_w \mid \mathbf{x}) \exp(r_\phi(\mathbf{x}, \mathbf{y}_w))/Z_\phi(\mathbf{x})] \Rightarrow$$

$$\min_{\phi} \mathbb{E}_{(\mathbf{x}, \mathbf{y}_w) \sim \mathcal{D}} \left[ -r_\phi(\mathbf{x}, \mathbf{y}_w) + \log \sum_{\mathbf{y}} \pi_{\mathrm{ref}}(\mathbf{y} \mid \mathbf{x}) \exp \left( r_\phi(\mathbf{x}, \mathbf{y}) \right) \right]. \tag{11}$$

There are several options for sampling from the reference distribution $\pi_{\text{ref}}(\mathbf{y} \mid \mathbf{x})$. A choice that simplifies the above expression and yields RLHF in practice is $\pi_{\text{ref}}(\mathbf{y} \mid \mathbf{x}) = \frac{1}{2}\mathbb{I}(\mathcal{Y} = \mathbf{y}_l) + \frac{1}{2}\mathbb{I}(\mathcal{Y} = \mathbf{y}_w)$. In this case, the sample-based approximation of the second term gives us:

$$\min_{\boldsymbol{\phi}} \mathbb{E}_{(\mathbf{x},\mathbf{y}_w,\mathbf{y}_l)\sim\mathcal{D}} \Big[ -r_{\boldsymbol{\phi}}(\mathbf{x},\mathbf{y}_w) + \log\big( \exp(r_{\boldsymbol{\phi}}(\mathbf{x},\mathbf{y}_w)) + \exp(r_{\boldsymbol{\phi}}(\mathbf{x},\mathbf{y}_l))\big)\Big]$$

$$= \mathbb{E}_{(\mathbf{x},\mathbf{y}_w,\mathbf{y}_l)\sim\mathcal{D}} \Big[ -\log\sigma\big(r_{\boldsymbol{\phi}}(\mathbf{x},\mathbf{y}_w) - r_{\boldsymbol{\phi}}(\mathbf{x},\mathbf{y}_l)\big)\Big]. \tag{12}$$

One can note that the above imitation learning loss over energy-based policy is exactly the same as the reward loss based on BT assumption in Equation (3) in RLHF. By optimizing this loss function, we can directly obtain the optimal energy-based policy in Equation (9). Unfortunately, even if we use the estimate $r_{\boldsymbol{\phi}}$, it is still expensive to estimate the partition function $Z_{\boldsymbol{\phi}}(\mathbf{x})$, making this representation difficult to use in practice and significantly higher inference cost (Rafailov et al., 2024). To address this problem, we can utilize the reverse knowledge distillation which distills the optimal policy in Equation (9) into a analytical policy by using the following reverse KL divergence, which allows the final policy $\pi_{\boldsymbol{\theta}}$ to require only a single sample during the inference time:

$$\min_{\boldsymbol{\theta}} \text{KL}\Big( \pi_{\boldsymbol{\theta}}(\mathbf{y} \mid \mathbf{x})\|\pi_{\text{ref}}(\mathbf{y} \mid \mathbf{x})\exp(r_{\boldsymbol{\phi}}(\mathbf{x},\mathbf{y})/\beta)/Z_{\boldsymbol{\phi}}(\mathbf{x})\Big), \tag{13}$$

where $\beta$ is the temperature hyperparameter in distillation process. This gives the following objective function after removing multiplicative and additive constants:

$$\mathcal{L}(\boldsymbol{\theta}) = -\mathbb{E}_{\pi_{\boldsymbol{\theta}}(\mathbf{y}|\mathbf{x})}\big[r_{\boldsymbol{\phi}}(\mathbf{x},\mathbf{y})\big] + \beta\text{KL}\big(\pi_{\boldsymbol{\theta}}(\mathbf{y} \mid \mathbf{x})\|\pi_{\text{ref}}(\mathbf{y} \mid \mathbf{x})\big). \tag{14}$$

One can observe that this distillation objective exactly corresponds to the RL objective in Equation (2).

In summary, we provide two key insights: (i) Reward learning in RLHF is equivalent to an imitation learning problem against the chosen responses, achieved by minimizing the forward KL divergence between $\pi_{\text{chosen}}$ and $\pi_{\boldsymbol{\phi}}$ based on the EBMs shown in Equation (12). (ii) The RL step in RLHF can be interpreted as a reverse knowledge distillation process, where the imitated policy $\pi_{\boldsymbol{\phi}}$, based on EBMs, is distilled into a final analytical policy $\pi_{\boldsymbol{\theta}}$ by minimizing the reverse KL divergence in Equation (13), with the temperature determining the level of KL regularization. Formally, we have:

**Proposition 4.1.** *Suppose the chosen response distribution $p(\mathbf{y} \mid \mathbf{x})$, the EBM $\pi_{\boldsymbol{\phi}}(\mathbf{y} \mid \mathbf{x})$, and the model $\pi_{\boldsymbol{\theta}}(\mathbf{y} \mid \mathbf{x})$. KL-regularized RLHF with $\beta = 1$ can be viewed as the following problem:*

$$\min_{\pi_{\boldsymbol{\theta}}} \text{KL}(\pi_{\boldsymbol{\theta}} \| \pi_{\boldsymbol{\phi}}^*) \quad \text{s.t.} \quad \pi_{\boldsymbol{\phi}}^* = \arg\min_{\pi_{\boldsymbol{\phi}}} \text{KL}(\pi_{\text{chosen}} \| \pi_{\boldsymbol{\phi}}), \tag{15}$$

*where $\pi_{\text{chosen}}(\mathbf{y} \mid \mathbf{x}) = \pi_{\boldsymbol{\phi}}(\mathbf{y} \mid \mathbf{x}) = \pi_{\boldsymbol{\theta}}(\mathbf{y} \mid \mathbf{x})$ is the equilibrium.*

Thus, conducting imitation learning on the chosen response corresponds to solving a standard KL-regularized RLHF problem. Additionally, we observe that the upper-level objective essentially optimizes a reverse KL (RKL) divergence, $\text{KL}(\pi_{\boldsymbol{\theta}} \| \pi_{\text{chosen}})$, given that $\pi_{\boldsymbol{\phi}}^* = \pi_{\text{chosen}}$, which is the optimum achieved by the lower-level objective.

An interesting question is why SFT, which directly optimizes the forward KL (FKL) $\text{KL}(\pi_{\text{chosen}} \| \pi_{\boldsymbol{\theta}})$ in Equation (5), performs worse than RLHF for alignment. Theoretically, minimizing SFT and RLHF should lead to the same optimal solution $\pi_{\boldsymbol{\theta}}$. However, achieving this in practice requires full data coverage and infinite computations, conditions that are rarely met. Consequently, in practical settings, minimizing either KL divergence results in learned policies with distinct properties, as discussed in (Murphy, 2012; Tajwar et al., 2024). Specifically, FKL $\text{KL}(\pi_{\text{chosen}} \| \pi_{\boldsymbol{\theta}})$ promotes mass-covering behavior, whereas RKL $\text{KL}(\pi_{\boldsymbol{\theta}} \| \pi_{\text{chosen}})$ encourages mode-seeking behavior (Tajwar et al., 2024; Nachum et al., 2016; Agarwal et al., 2019). Mass-covering encourages assigning equal probability to all responses in the dataset, leading to an overestimation of the long tail of the target distribution, while mode-seeking concentrates the probability mass on specific high-reward regions. Thus, alignment focuses on generating a certain subset of high-reward responses, which is more effectively achieved by minimizing reverse KL, as theoretically shown by (Tajwar et al., 2024; Ji et al., 2024).

## 4.2 DIRECT IMITATION LEARNING

In the last section, we revisit RLHF from the perspective of imitation learning. Our analysis explicitly suggests that RLHF is essentially optimized to align closely with the distribution of the chosen

Table 1: Summary of the variants of `DIL` with different $h$-functions for Bregman divergence: $\mathcal{L}_{\text{DIL}}(\boldsymbol{\theta}) = \mathbb{E}_{\pi_{\text{chosen}}(\mathbf{y}|\mathbf{x})}[\ell_1(f_{\boldsymbol{\theta}})] + \mathbb{E}_{\pi_{\text{rejected}}(\mathbf{y}|\mathbf{x})}[\ell_{-1}(f_{\boldsymbol{\theta}})]$ as a function of log ratio $f_{\boldsymbol{\theta}} = \log(\pi_{\boldsymbol{\theta}}(\mathbf{y}\mid\mathbf{x})/\pi_{\text{ref}}(\mathbf{y}\mid\mathbf{x}))$.

| $h$-Bregman Density Ratio Estimation | $h$-function | $\ell_1(f_{\boldsymbol{\theta}})$ | $\ell_{-1}(f_{\boldsymbol{\theta}})$ |
|---|---|---|---|
| LSIF (Kanamori et al., 2009) | $h(r) = (r-1)^2/2$ | $-e^{f_{\boldsymbol{\theta}}}$ | $\frac{1}{2}e^{2f_{\boldsymbol{\theta}}}$ |
| BCE (Hastie et al., 2009) | $h(r) = r\log r - (r+1)\log(r+1)$ | $\log(1+e^{-f_{\boldsymbol{\theta}}})$ | $\log(1+e^{f_{\boldsymbol{\theta}}})$ |
| UKL (Nguyen et al., 2010) | $h(r) = r\log r - r$ | $-f_{\boldsymbol{\theta}}$ | $e^{f_{\boldsymbol{\theta}}}$ |

responses. The sample-based approximation of EBMs in `RLHF` results in a reward loss similar to the BT model, as shown in Equation (12). However, the BT assumption may not always hold true, as discussed in (Azar et al., 2024; Munos et al., 2023; Sun & van der Schaar, 2024). Based on these insights, we propose a novel alignment method, `DIL`, without the BT assumption. We directly formulate the objective of imitation learning as minimizing the reverse KL divergence between $\pi_{\boldsymbol{\theta}}$ and the unknown distribution of the chosen response $\pi_{\text{chosen}}$ (Kostrikov et al., 2019; Fu et al., 2018):

$$\min_{\boldsymbol{\theta}} \mathcal{L}_{\text{DIL}}(\boldsymbol{\theta}) = \text{KL}\Big(\pi_{\boldsymbol{\theta}}(\mathbf{y}\mid\mathbf{x})\|\pi_{\text{chosen}}(\mathbf{y}\mid\mathbf{x})\Big) = \mathbb{E}_{\pi_{\boldsymbol{\theta}}(\mathbf{y}|\mathbf{x})}\Big[\log\Big(\pi_{\boldsymbol{\theta}}(\mathbf{y}\mid\mathbf{x})/\pi_{\text{chosen}}(\mathbf{y}\mid\mathbf{x})\Big)\Big], \quad (16)$$

where we minimize RKL divergence, rather than FKL divergence as in `SFT`, as shown in Equation (5).

However, mode-seeking with reverse KL divergence is generally challenging. Directly optimizing Equation (16) does not effectively leverage chosen preference data, particularly because the data policy $\pi_{\text{chosen}}$ is unknown. In the RL literature, these challenges have been addressed through adversarial training (Ho & Ermon, 2016; Fu et al., 2018). However, these methods require learning a reward function using complex and unstable adversarial training, which is impractical for large models. In this paper, we propose a straightforward alternative that leverages preference data without learning a reward function via adversarial training. We reformulate the `DIL` objective as follows:

$$\max_{\boldsymbol{\theta}} \mathbb{E}_{\pi_{\boldsymbol{\theta}}(\mathbf{y}|\mathbf{x})}\Big[\log\frac{\pi_{\text{chosen}}(\mathbf{y}\mid\mathbf{x})}{\pi_{\text{ref}}(\mathbf{y}\mid\mathbf{x})} - \log\frac{\pi_{\boldsymbol{\theta}}(\mathbf{y}\mid\mathbf{x})}{\pi_{\text{ref}}(\mathbf{y}\mid\mathbf{x})}\Big] =$$
$$\mathbb{E}_{\pi_{\boldsymbol{\theta}}(\mathbf{y}|\mathbf{x})}\Big[\log r(\mathbf{x},\mathbf{y})\Big] - \text{KL}\big(\pi_{\boldsymbol{\theta}}(\mathbf{y}\mid\mathbf{x})\|\pi_{\text{ref}}(\mathbf{y}\mid\mathbf{x})\big), \quad (17)$$

where $r(\mathbf{x},\mathbf{y}) \triangleq \frac{\pi_{\text{chosen}}(\mathbf{y}|\mathbf{x})}{\pi_{\text{ref}}(\mathbf{y}|\mathbf{x})}$ can be viewed as an auxiliary reward function. Equations (16) and (17) are equivalent by adding and subtracting the same term of $\log \pi_{\text{ref}}(\mathbf{y}\mid\mathbf{x})$ in the expectation.

Interestingly, we find that even when only preference data is available, this objective takes a form similar to the `RLHF` objective in Equation (2). The primary difference lies in the reward being the estimated log density ratio, which is often not readily accessible in real-world applications. Optimizing this objective, which involves the density ratio $r(\mathbf{x},\mathbf{y})$, is not straightforward. In the next section, we demonstrate how to efficiently optimize it by effectively utilizing offline human preference data.

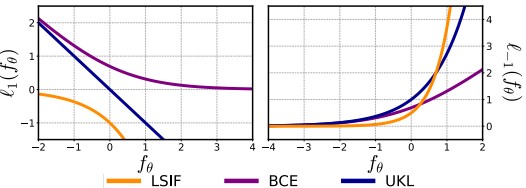

Figure 1: The illustration of different losses (LSIF, BCE, and UKL), as shown in Table 1.

### 4.3 DENSITY RATIO REWARD ESTIMATION

Before delving into the problem in Equation (17), we first describe how to calculate the auxiliary reward function in terms of the density ratio. In the tabular setting, we can directly compute $\pi_{\text{ref}}(\mathbf{y}\mid\mathbf{x})$ and $\pi_{\text{chosen}}(\mathbf{y}\mid\mathbf{x})$. However, in a high-dimensional language domain, estimating the densities separately and then calculating their ratio hardly works well due to error accumulation. In this paper, we choose to directly estimate the density ratio $\pi_{\text{chosen}}(\mathbf{y}\mid\mathbf{x})/\pi_{\text{ref}}(\mathbf{y}\mid\mathbf{x})$ based on the Bregman divergence (Sugiyama et al., 2012). Suppose $r^*(\mathbf{x},\mathbf{y}) = \pi_{\text{chosen}}(\mathbf{y}\mid\mathbf{x})/\pi_{\text{ref}}(\mathbf{y}\mid\mathbf{x})$ is the target density ratio to be estimated with a parameterized discriminator $r_\phi$. Then, we have:

$$\min_{\phi} \text{D}_h(r^*\|r_\phi) = \sum_{\mathbf{y}} \pi_{\text{ref}}(\mathbf{y}\mid\mathbf{x})\text{B}_h(r^*(\mathbf{x},\mathbf{y})\|r_\phi(\mathbf{x},\mathbf{y}))$$
$$= \sum_{\mathbf{y}} \pi_{\text{ref}}(\mathbf{y}\mid\mathbf{x})\Big(h\big(r^*(\mathbf{x},\mathbf{y})\big) - h\big(r_\phi(\mathbf{x},\mathbf{y})\big) - \partial h\big(r_\phi(\mathbf{x},\mathbf{y})\big)\big(r^*(\mathbf{x},\mathbf{y}) - r_\phi(\mathbf{x},\mathbf{y})\big)\Big), \quad (18)$$

where $\text{B}_h$ is the data-level Bregman divergence. For a twice continuously differentiable convex function $h$ with a bounded derivative $\partial h$, this divergence quantifies the discrepancy between two

density-ratios. Subtracting a constant $\sum_{\mathbf{y}} \pi_{\text{ref}}(\mathbf{y} \mid \mathbf{x}) h(r^*(\mathbf{x}, \mathbf{y}))$, we obtain (up to a constant):

$$\sum_{\mathbf{y}} \pi_{\text{ref}}(\mathbf{y} \mid \mathbf{x}) \Big[ \partial h\big(r_\phi(\mathbf{x}, \mathbf{y})\big) r_\phi(\mathbf{x}, \mathbf{y}) - h\big(r_\phi(\mathbf{x}, \mathbf{y})\big) \Big] - \sum_{\mathbf{y}} \pi_{\text{chosen}}(\mathbf{y} \mid \mathbf{x}) \Big[ \partial h\big(r_\phi(\mathbf{x}, \mathbf{y})\big) \Big]. \quad (19)$$

A few non-exhaustive examples of the Bregman divergence are Least-Squared Importance Fitting (LSIF) (Kanamori et al., 2009), Binary Cross Entropy (BCE) (Hastie et al., 2009), and the unbounded Kullback-Leibl (UKL) (Nguyen et al., 2010). For example, LSIF defines $h_{\text{LSIF}} = (r - 1)^2/2$, which results in the following instance of Bregman divergence on the density ratio:

$$\min_{\phi} \mathrm{D}_{h_{\text{LSIF}}}(r^* \| r_\phi) = \sum_{\mathbf{y}} \frac{1}{2} \pi_{\text{ref}}(\mathbf{y} \mid \mathbf{x}) r_\phi^2(\mathbf{x}, \mathbf{y}) - \pi_{\text{chosen}}(\mathbf{y} \mid \mathbf{x}) r_\phi(\mathbf{x}, \mathbf{y}) \quad (20)$$

In this case, sample-based approximation of Equation (20) leads to the following loss function:

$$\mathcal{L}(\phi; \mathcal{D}) = \mathbb{E}_{(\mathbf{x}, \mathbf{y}_w, \mathbf{y}_l) \sim \mathcal{D}} \Big[ \frac{1}{2} r_\phi^2(\mathbf{x}, \mathbf{y}_l) - r_\phi(\mathbf{x}, \mathbf{y}_w) \Big], \quad (21)$$

Here, we use the set of rejected responses $\mathbf{y}_l \sim \pi_{\text{ref}}(\mathbf{y} \mid \mathbf{x})$ to approximate the expectations under $\pi_{\text{ref}}(\mathbf{y} \mid \mathbf{x})$. It is acceptable to use the set of rejected responses $y_l$ from the preference dataset $\mathcal{D}$ to approximate the expectations. We can even make use of both chosen responses and rejected responses to approximate these expectations. However, since our goal is to decrease the likelihood of rejected responses, we choose to use the rejected responses to approximate the expectations, and we find it empirically works well. Intuitively, the first term pushes the model to decrease the density ratio of the rejected response, while the second term increases the density ratio of the chosen response. In addition, this direct estimation approach with $h$ Bregman divergence suggests a divergence family for density ratio estimation as shown in Table 1; see Appendix A for further discussion of other $h$ functions such as BCE (Hastie et al., 2009) and UKL (Nguyen et al., 2010). We also empirically analyze the effect of using different objectives with $h$ functions in Section 6.3.

## 4.4 OPTIMIZATION

So far, we have observed that the RL-style objective in Equation (17), combined with density ratio estimation in Equation (21), can effectively leverage the preference dataset for imitation learning. However, this two-step process is complex and unstable, requiring first the fitting of a reward model to estimate the density ratio and then fine-tuning the language model policy using the RL-style objective in Equation (17). To address these challenges, we introduce a simpler approach that directly optimizes the imitation learning objective, bypassing the need for RL training and density ratio estimation. The core idea lies in a specialized parameterization of the density ratio reward, which allows for the direct extraction of the optimal policy, eliminating the need for an RL loop (Rafailov et al., 2024). Notably, the optimal policy in Equation (17) has a closed-form solution, as shown by (Rafailov et al., 2024):

$$\pi^*(\mathbf{y} \mid \mathbf{x}) = \frac{1}{Z(\mathbf{x})} \pi_{\text{ref}}(\mathbf{y} \mid \mathbf{x}) \exp\big(\log r^*(\mathbf{x}, \mathbf{y})\big), \quad (22)$$

where $Z(\mathbf{x}) = \sum_{\mathbf{y}} \pi_{\text{ref}}(\mathbf{y}|\mathbf{x}) \exp\big(\log r^*(\mathbf{x}, \mathbf{y})\big) = \sum_{\mathbf{y}} \pi_{\text{chosen}}(\mathbf{y}|\mathbf{x}) = 1$, meaning that the optimal $\pi^*(\mathbf{y}|\mathbf{x})$ is forced to be self-normalized! This characteristic, determined by the reward definition in Equation (17), is super beneficial as it allows our imitation learning to theoretically generalize to a broader class of loss functions beyond the pairwise BT preference model used in DPO. Taking the logarithm of both sides of Equation (22) and then with some algebra, we obtain the following:

$$\log \frac{\pi^*(\mathbf{y} \mid \mathbf{x})}{\pi_{\text{ref}}(\mathbf{y} \mid \mathbf{x})} = \log r^*(\mathbf{x}, \mathbf{y}), \quad (23)$$

where $r^*(\mathbf{x}, \mathbf{y})$ is the density ratio estimated by Equation (21) on the preference dataset. Since the optimal density ratio is now represented in terms of the optimal policy, rather than the discriminator model, we can explicitly derive the following maximum likelihood objective for a parameterized policy over the preference dataset (Rafailov et al., 2024). Analogous to the approach used for density ratio estimation and utilizing a change of variables, we can formalize our DIL objective as follows:

$$\mathcal{L}_{\text{DIL}}(\boldsymbol{\theta}; \mathcal{D}) = \mathbb{E}_{(\mathbf{x}, \mathbf{y}_w, \mathbf{y}_l) \sim \mathcal{D}} \Big[ -\frac{\pi_{\boldsymbol{\theta}}(\mathbf{y}_w \mid \mathbf{x})}{\pi_{\text{ref}}(\mathbf{y}_w \mid \mathbf{x})} + \frac{1}{2} \big( \frac{\pi_{\boldsymbol{\theta}}\big(\mathbf{y}_l \mid \mathbf{x}\big)}{\pi_{\text{ref}}\big(\mathbf{y}_l \mid \mathbf{x}\big)} \big)^2 \Big], \quad (24)$$

where we directly fit an implicit density ratio in Equation (21) using an alternative parameterization in Equation (23). Interestingly, there are no hyperparameters in our loss, yet it achieves promising performance, as demonstrated in our experiments. Since our procedure is equivalent to fitting a reparameterized density ratio estimation model, it theoretically conducts imitation learning by minimizing RKL divergence against the unknown distribution of the chosen response. Table 1 shows a family of objectives that meet the definition of Bregman divergence.

## 4.5 DISCUSSION: DPO IS A SPECIAL CASE OF DIL

In this section, we show that DPO can be also viewed as a special case of our framework by using contrastive predictive coding (CPC) (also as known as InfoNCE) (Oord et al., 2018) for density ratio estimation. Given the prompt distribution $p(\mathbf{x})$ and the conditional distribution of the chosen response $\pi_{\text{chosen}}(\mathbf{y} \mid \mathbf{x})$, we sample $\mathbf{x} \sim p(\mathbf{x})$, $\mathbf{y}_w \sim \pi_{\text{chosen}}(\mathbf{y} \mid \mathbf{x})$, and $\mathbf{y}_l \sim \pi_{\text{ref}}(\mathbf{y} \mid \mathbf{x})$. CPC optimizes:

$$\mathcal{L}_{\text{CPC}}(\boldsymbol{\phi}; \mathcal{D}) = -\mathbb{E}_{(\mathbf{x}, \mathbf{y}_w, \mathbf{y}_l) \sim \mathcal{D}} \Big[ \log \frac{\exp(f_{\boldsymbol{\phi}}(\mathbf{x}^\top \mathbf{y}_w)/\beta)}{\exp(f_{\boldsymbol{\phi}}(\mathbf{x}^\top \mathbf{y}_w/\beta)) + \exp(f_{\boldsymbol{\phi}}(\mathbf{x}^\top \mathbf{y}_l)/\beta)} \Big], \tag{25}$$

where $f_{\boldsymbol{\phi}} : \mathcal{X} \times \mathcal{Y} \mapsto \mathbb{R}$ is a parametric critic function. The optimal critic for this CPC with one negative sample satisfies the following (Zheng et al., 2024; Ma & Collins, 2018; Oord et al., 2018):

$$f^*(\mathbf{x}, \mathbf{y})/\beta = \log \frac{\pi_{\text{chosen}}(\mathbf{y} \mid \mathbf{x})}{\pi_{\text{ref}}(\mathbf{y} \mid \mathbf{x}) c(\mathbf{x})} = \log r^*(\mathbf{x}, \mathbf{y}) - \log c(\mathbf{x}), \tag{26}$$

where $c(\mathbf{x})$ is a function (Oord et al., 2018; Zheng et al., 2024), that depends on $\mathbf{x}$ but not $\mathbf{y}$. Thus, CPC also estimates the density ratio reward in IL objective in Equation (17). Similar to Section 4.4, by using the closed-form optimal policy in Equation (22) and using a change of variables, we have:

$$\mathcal{L}_{\text{DIL}}(\boldsymbol{\theta}; \mathcal{D}) = -\mathbb{E}_{(\mathbf{x}, \mathbf{y}_w, \mathbf{y}_l) \sim \mathcal{D}} \Big[ \log \sigma \big( \beta \log \frac{\pi_{\boldsymbol{\theta}}(\mathbf{y}_w \mid \mathbf{x})}{\pi_{\text{ref}}(\mathbf{y}_w \mid \mathbf{x})} - \beta \log \frac{\pi_{\boldsymbol{\theta}}(\mathbf{y}_l \mid \mathbf{x})}{\pi_{\text{ref}}(\mathbf{y}_l \mid \mathbf{x})} \big) \Big], \tag{27}$$

which is exactly the same objective as the well-known DPO. Thus, our framework enables us to reinterpret DPO. Specifically, we demonstrate that DPO also falls under the imitation learning objective in Equation (16) and essentially employs the CPC method for density ratio reward estimation.

## 5 EXPERIMENTS

**Datasets.** We evaluate DIL on widely used datasets: the UltraFeedback Binarized dataset (Cui et al., 2023; Tunstall et al., 2023), the Reddit TL;DR summarization dataset (Völske et al., 2017), and the Anthropic-HH dataset (Bai et al., 2022). The details of these datasets are provided in Appendix B.1.

**Tasks and Evaluation.** Following previous work (Rafailov et al., 2024; Tunstall et al., 2023), we evaluate methods fine-tuned on the UltraFeedback Binarized dataset across tasks on the Open LLM Leaderboard (Gao et al., 2023). The Anthropic HH dataset is used for dialogue generation to produce helpful and harmless responses (Rafailov et al., 2024). For summarization, we use the Reddit TL;DR dataset. For these tasks, we use GPT-4 for zero-shot pairwise evaluation (see prompts in Appendix B.2). The task and evaluation details are also provided in Appendix B.2.

**Models.** For summarization and dialogue generation tasks, we use Pythia-2.8b (Biderman et al., 2023) as our base model, with the model after SFT serving as a reference model, following (Rafailov et al., 2024). For fine-tuning on the UltraFeedback Binarized dataset, we use Mistral-7B-Base (Tunstall et al., 2023) and Llama3-8b-SFT used in (Meng et al., 2024) as our base models.

**Baselines and Implementation.** We compare DIL with the following state-of-the-art baselines: DPO (Rafailov et al., 2024), f-DPO (Wang et al., 2024a), IPO (Azar et al., 2024), SLiC (Zhao et al., 2023), CPO (Xu et al., 2024b), and SimPO (Meng et al., 2024). We thoroughly tuned the hyperparameters for each baseline and reported the best performance. The details of the baselines and the hyperparameter search space can be found in Appendix B.3. The density ratio in Section 4.3 is estimated through optimization toward the Bregman divergence. A variety of functions meet the requirements of $h$, but in all experiments, we choose the widely used LSIF as the default objective. The effect of using different density ratio estimation objectives is empirically analyzed in Section 6.3.

Table 2: Evaluation results on various tasks from the Huggingface Open Leaderboard v1 and v2. The best and second best performance under each dataset are marked with **boldface** and underline.

| Model (↓) / Benchmark (→) | | MMLU-PRO | BBH | MUSR | MATH | GSM8K | ARC |
|---|---|---|---|---|---|---|---|
| | SFT | **27.58** | 41.26 | 41.93 | 2.34 | 28.13 | 58.28 |
| Mistral-7B-Base | DPO (Rafailov et al., 2024) | 26.73 | 43.27 | 43.65 | 1.36 | 21.76 | 61.26 |
| | SLiC (Zhao et al., 2023) | 26.52 | 42.33 | 33.74 | 1.38 | 33.74 | 55.38 |
| | f-DPO (Wang et al., 2024a) | 25.96 | 42.39 | 37.82 | 1.27 | 23.18 | 62.01 |
| | IPO (Azar et al., 2024) | 25.87 | 40.59 | 42.15 | 1.25 | 27.14 | 60.84 |
| | KTO (Ethayarajh et al., 2024) | 27.35 | 42.37 | 43.17 | 2.34 | **36.58** | 62.39 |
| | CPO (Xu et al., 2024b) | 27.04 | 42.05 | 42.15 | 2.15 | 33.06 | 57.00 |
| | SimPO (Meng et al., 2024) | 27.13 | 42.94 | 39.68 | 2.49 | 22.21 | 62.63 |
| | DIL w/ LSIF | 27.44 | **43.59** | **44.05** | **2.95** | 32.19 | **63.31** |
| | SFT | 31.00 | 46.16 | 41.27 | 3.70 | 46.32 | 60.15 |
| LLama3-8B-Base | DPO (Rafailov et al., 2024) | 31.58 | 47.80 | 40.48 | 4.53 | 38.67 | 64.42 |
| | SLiC (Zhao et al., 2023) | 31.11 | 46.53 | 40.55 | 3.92 | 48.82 | 61.43 |
| | f-DPO (Wang et al., 2024a) | 30.85 | 47.55 | 40.39 | 4.37 | 39.55 | 62.85 |
| | IPO (Azar et al., 2024) | 30.18 | 46.78 | 39.58 | 4.02 | 22.67 | 62.88 |
| | KTO (Ethayarajh et al., 2024) | 31.16 | 47.92 | 40.24 | 4.13 | 38.99 | 63.17 |
| | CPO (Xu et al., 2024b) | 30.95 | 47.17 | 41.59 | 4.25 | 46.93 | 61.69 |
| | SimPO (Meng et al., 2024) | 31.61 | 48.38 | 40.08 | 4.23 | 31.54 | 65.19 |
| | DIL w/ LSIF | **32.22** | **48.78** | **42.75** | **4.68** | **48.98** | **65.37** |

Table 3: Win rates computed by GPT-4 against the SFT generated response and the chosen responses on the TL;DR summarization and Anthropic-HH datasets on Pythia-2.8b. The best and second best performance under each dataset are marked with **boldface** and underline, respectively.

| Dataset (→) | TL;DR Summarization | | | Anthropic-HH | | |
|---|---|---|---|---|---|---|
| Method (↓) / Metric (→) | vs SFT | vs Chosen | Average | vs SFT | vs Chosen | Average |
| DPO (Rafailov et al., 2024) | 71.22 | 57.58 | 64.40 | 69.32 | 59.35 | 64.34 |
| SLiC (Zhao et al., 2023) | 68.61 | 55.72 | 62.17 | 65.52 | 57.71 | 61.62 |
| f-DPO (Wang et al., 2024a) | 66.19 | 51.37 | 58.78 | 60.21 | 52.38 | 56.30 |
| IPO (Azar et al., 2024) | 72.17 | 56.51 | 64.34 | 63.19 | 55.12 | 59.16 |
| CPO (Xu et al., 2024b) | 73.13 | 58.89 | 66.01 | 72.30 | 63.39 | 67.86 |
| SimPO (Meng et al., 2024) | 69.71 | 54.38 | 62.05 | 67.85 | 57.51 | 62.68 |
| DIL w/ LSIF | **75.47** | **60.25** | **67.86** | **73.32** | **65.02** | **69.17** |

## 6 EXPERIMENTAL RESULTS

### 6.1 PERFORMANCE COMPARISON ON BENCHMARKS

In this section, as shown in Table 2, we compare the performance of DIL against other alignment methods on Open LLM Leaderboard. Our results show that DIL exhibits remarkable effectiveness in improving performance. Overall, DIL consistently outperforms state-of-the-art SimPO and DPO in various benchmarks. For instance, on

Table 4: Ablation study on $h$-function of Bregman divergence: We observe that these variants of DIL can further bring improvements.

| Model (↓) / Benchmark (→) | | BBH | MUSR | MATH | GSM8K |
|---|---|---|---|---|---|
| Mistral-7B Base | DIL w/ LSIF | 43.59 | 44.05 | **2.95** | 32.19 |
| | DIL w/ UKL | 43.92 | **45.11** | 2.04 | 30.71 |
| | DIL w/ BCE | **45.13** | 43.92 | 2.79 | **33.13** |
| LLama3-8B Base | DIL w/ LSIF | 48.78 | 42.75 | 4.68 | 48.98 |
| | DIL w/ UKL | **49.71** | 43.01 | 4.98 | **50.95** |
| | DIL w/ BCE | 48.96 | **47.35** | **5.06** | 49.36 |

LLama3, the improvements are notable on the Math benchmarks, with relative gains exceeding 7.5% over SimPO. Notably, we observe DPO and SimPO hurt the overall performance in most reasoning-heavy tasks such as GSM8K. This indicates that both SimPO and DPO might not be suitable to improve reasoning abilities, which is consistent with findings in concurrent works (Pal et al., 2024; Meng et al., 2024). In contrast, DIL shows clear improvements in both the Mistral and LLama3

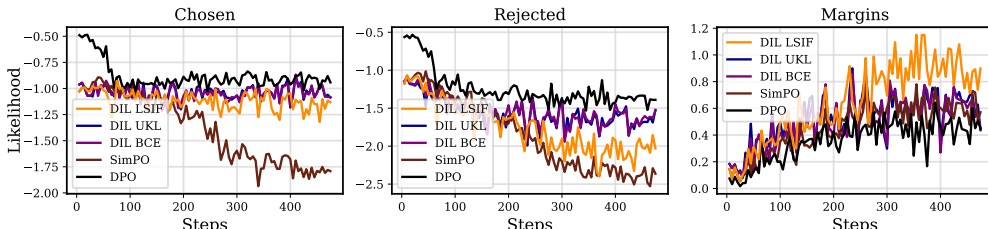

Figure 2: The training dynamics of `DIL` variants, DPO and `SimPO` on Mistral show that `DIL` exhibits the smallest decline in chosen likelihoods, while still increasing the likelihood margins between rejected and chosen responses, compared to `SimPO` and DPO. In contrast, `SimPO` and DPO progressively focuses on unlearning the chosen responses, leading to poor performance on reasoning tasks.

models. These findings underscore the effectiveness of `DIL`. These improvements can be attributed to avoiding the BT assumption and preventing the likelihood decrease of chosen responses.

## 6.2 PERFORMANCE COMPARISON WITH HUMAN PREFERENCES

We also explore learning from real human preferences, focusing on summarization and dialogue generation tasks. Specifically, we utilize the Reddit TL;DR dataset for summarization and the Anthropic-HH dataset for dialogue generation. We employ Pythia-2.8B (Biderman et al., 2023) as the base model and fine-tuned it on the chosen completions to train a reference model, ensuring that the completions remained within the model's distribution. Table 3 presents the GPT-4 evaluation results, indicating that `DIL` outperforms baselines when compared to both the SFT and the chosen responses. Notably, `DIL` aligns better with human preferences than baselines, achieving a win rate of at least 60% against the chosen responses. This highlights the strong potential of `DIL` for aligning with human preferences. Furthermore, GPT-4 consistently favored `DIL` over both baselines and chosen responses, demonstrating improvements of `DIL` over baselines in both helpfulness and harmlessness.

## 6.3 FURTHER ANALYSIS

**Generalization to other objectives.** As mentioned in Section 4.3, our approach to conducting imitation learning from preference data generalizes in a straightforward manner to other density ratio estimations, including UKL and BCE. Table 4 shows the comparison on UltraFeedback. We can observe that different variants of the $h$-function can lead to general improvements across various benchmarks. Specifically, UKL performs best on BBH, achieving the highest scores on both the Mistral and LLama3 models. BCE achieves a significant improvement on MUSR, with a notable 7.27% increase. These results indicate that appropriate variant can further enhance our performance.

**Training Dynamics.** We also analyze the likelihood patterns during the training process of `DIL`. Figure 2 illustrates the likelihood trends of `SimPO` and `DIL` on UltraFeedback. We observe that the likelihood of rejected responses continues to decrease, while the margin between chosen and rejected responses steadily increases. However, for DPO and `SimPO`, the likelihood of chosen responses drops below zero and continues to decline. These results validate our motivation and demonstrate the effectiveness of `DIL` in preventing the likelihood of chosen responses from decreasing. This also explains why `DIL` generally enhances downstream task performance, particularly in reasoning-heavy tasks such as math, as shown in Table 2.

## 7 CONCLUSION

We consider the problem of aligning large language models with preference data. We provide a novel perspective on imitation learning for the alignment problem, and demonstrate `RLHF/DPO` essentially conduct imitation learning on the distribution of chosen response with reverse KL divergence. Building upon this connection, we propose `DIL`, which directly optimizes the imitation learning objective based on Bregman divergence and can more effectively preserve reasoning abilities than baselines while aligning with human preferences. Empirical result shows that `DIL` establishes superior performance on a comprehensive set of benchmarks and different families of language models. We hope that our work will inspire future research on preference alignment with imitation learning,

ACKNOWLEDGMENT

The work of Vasant G Honavar and Teng Xiao was supported in part by grants from the National Science Foundation (2226025, 2225824), the National Center for Advancing Translational Sciences, and the National Institutes of Health (UL1 TR002014).

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

## A    DETAILS OF METHODS FOR DENSITY RATIO ESTIMATION

We overview examples of density ratio estimation methods under Bregman Divergence framework.

**Least Squares Importance Fitting (LSIF).** LSIF (Kanamori et al., 2009) minimizes the squared error between a density ratio model $r$ and the true density ratio $r^*$ defined as follows:

$$\begin{aligned}
D_{h_{\text{LSIF}}}(r^* \| r_\phi) &= \mathbb{E}_{\pi_{\text{ref}}}[(r_\phi(\mathbf{x}, \mathbf{y}) - r^*(\mathbf{x}, \mathbf{y})^2] \\
&= \mathbb{E}_{\pi_{\text{ref}}}[(r^*(\mathbf{x}, \mathbf{y}))^2] - 2\mathbb{E}_{\pi_{\text{chosen}}}[r_\phi(\mathbf{x}, \mathbf{y})] + \mathbb{E}_{\pi_{\text{ref}}}[(r_\phi(\mathbf{x}, \mathbf{y}))^2],
\end{aligned} \quad (28)$$

where the first term in the above equation is constant w.r.t $\phi$. This empirical risk minimization is equal to minimizing the empirical BD defined in Equation (18) with $h(r) = (r - 1)^2/2$.

**KL Importance Estimation Procedure (KLIEP).** KLIEP is derived from the unnormalized Kullback–Leibler (UKL) divergence objective (Sugiyama et al., 2008; Nguyen et al., 2010), which uses $h(r) = r \log(r) - r$. Ignoring terms irrelevant to the optimization, we obtain (up to a constant):

$$D_{h_{\text{KLIEP}}}(r^* \| r_\phi) = \mathbb{E}_{\pi_{\text{ref}}}\left[ r(\mathbf{x}, \mathbf{y}) \right] - \mathbb{E}_{\pi_{\text{chosen}}}\left[ \log\left( r(\mathbf{x}, \mathbf{y}) \right) \right]. \quad (29)$$

KLIEP is also known as solving a Lagrangian of the constrained problem with further imposing a constraint that the ratio model $r(\mathbf{x}, \mathbf{y})$ is non-negative and normalized as follows:

$$\max_r \mathbb{E}_{\pi_{\text{chosen}}}\left[ \log\left( r(\mathbf{x}, \mathbf{y}) \right) \right] \quad (30)$$

$$\text{s.t. } \mathbb{E}_{\pi_{\text{ref}}}\left[ r(\mathbf{x}, \mathbf{y}) \right] = 1 \text{ and } r(\mathbf{x}, \mathbf{y}) \geq 0 \text{ for all } (\mathbf{x}, \mathbf{y}). \quad (31)$$

**Binary Cross Entropy.**    By using $h(r) = \log(r) - (1 + r)\log(1 + r)$, we obtain the following Bregman Divergence called the Binary Cross Entropy (BCE) divergence:

$$D_{h_{\text{BCE}}}(r^* \| r_\phi) = -\mathbb{E}_{\pi_{\text{ref}}}\left[ \log\left( \frac{1}{1 + r(\mathbf{x}, \mathbf{y})} \right) \right] - \mathbb{E}_{\pi_{\text{chosen}}}\left[ \log\left( \frac{r(\mathbf{x}, \mathbf{y})}{1 + r(\mathbf{x}, \mathbf{y})} \right) \right].$$

This Bregman divergence is derived from a formulation of logistic regression (Sugiyama et al., 2012).

## B    EXPERIMENTAL DETAILS

### B.1    THE DETAILS OF DATASETS

**UltraFeedback Binarized** (Cui et al., 2023; Tunstall et al., 2023): This dataset[1] consists of 64k prompts, each paired with four completions generated by various open-source and proprietary models. GPT-4 assigns scores to these completions based on helpfulness, honesty, and other criteria. Binary preferences are then constructed by selecting the completion with the highest score as the chosen response, while one of the remaining three completions is randomly selected as the rejected response.

**Anthropic-HH** (Bai et al., 2022): The Anthropic Helpful and Harmless dialogue dataset[2] comprises 170k dialogues between humans and LLM assistants, designed for evaluating single-turn dialogue tasks. Each dialogue consists of a human query and two model responses rated for helpfulness and harmlessness. Following DPO (Rafailov et al., 2024), the chosen responses from this dataset were used during the supervised fine-tuning (SFT) phase.

**Reddit TL;DR Summarization** (Völske et al., 2017): This dataset[3] comprises Reddit forum posts curated for summarization tasks with associated preference labels. Following prior work (Stiennon et al., 2020), we use a filtered version of this dataset to train the SFT policy and leverage its preference labels during the subsequent alignment phase.

---

[1]https://huggingface.co/datasets/HuggingFaceH4/ultrafeedback_binarized
[2]https://huggingface.co/datasets/Anthropic/hh-rlhf
[3]https://huggingface.co/datasets/openai/summarize_from_feedback

Table 5: The hyperparameter search space for the baselines.

| Method | | Method | |
|---|---|---|---|
| DPO/f-DPO | $\beta \in [0.01, 0.05, 0.1]$ | IPO | $\tau \in [0.01, 0.1, 0.5, 1.0]$ |
| CPO | $\lambda = 1.0$ 
 $\beta \in [0.01, 0.05, 0.1]$ | SLiC | $\lambda \in [0.1, 0.5, 1.0, 10.0]$ 
 $\delta \in [0.1, 0.5, 1.0, 2.0]$ |
| KTO | $\lambda_l = \lambda_w = 1.0$ 
 $\beta \in [0.01, 0.05, 0.1]$ | SimPO | $\beta \in [2.0, 2.5]$ 
 $\gamma \in [0.3, 0.5, 1.0, 1.2, 1.4, 1.6]$ |

## B.2 THE DETAILS OF TASKS AND EVALUATION

This section introduces the benchmark for model evaluation. The model fine-tuned on the UltraFeedback Binarized dataset is evaluated following previous works (Rafailov et al., 2024; Tunstall et al., 2023) using the HuggingFace Open LLM Leaderboard v1[4] and v2[5] (Beeching et al., 2023; Fourrier et al., 2024). The evaluation includes MMUL-PRO, BBH, MUSR, MATH, GSM8k, and ARC. Models fine-tuned on the Anthropic-HH dataset follow the evaluation protocol outlined by (Rafailov et al., 2024), leveraging GPT-4 for zero-shot pairwise evaluation.

**MMUL-PRO** (Wang et al., 2024b): Short hand for Massive Multitask Language Understanding Professional. This dataset builds on the MMLU by incorporating more complex multiple-choice questions and undergoing rigorous expert review to enhance quality, difficulty and reduce data biases.

**BBH** (Suzgun et al., 2022): Short hand for Big Bench Hard. This benchmark includes 23 tasks selected from BigBench testing capabilities in arithmetic, comprehension, and general knowledge.

**MUSR** (Sprague et al., 2024): Short hand for Multistep Soft Reasoning. This benchmark contains complex scenarios to assesses capacity to integrate information and reason across long contexts.

**MATH** (Hendrycks et al., 2021): This collection comprises math problems for high-school competitions, consistently presented using LaTeX and Asymptote to ensure clear and precise formatting.

**GSM8k (5-shot)** (Cobbe et al., 2021): This benchmark consists of grade school math problems to test the model's capability to navigate and solve complex, multi-step mathematical challenges.

**ARC (25-shot)** (Clark et al., 2018): Short hand for AI2 Reasoning Challenge. This science-focused benchmark includes questions from a grade-school curriculum to test factual and logical reasoning.

**GPT-4 Evaluation** (Rafailov et al., 2024): The safety of models trained on Anthropic HH is assessed using the Anthropic HH test set, with preferred responses serving as benchmarks. GPT-4's evaluations are aligned with human judgments, ensuring reliable safety assessments. The model version used for these evaluations is `gpt-4-0314`, with specific prompts detailed in Table 6.

## B.3 IMPLEMENTATION DETAILS

**Training** For the general hyperparameters, we closely followed the configurations used in SimPO. Specifically, during the SFT stage, we applied a learning rate of $2 \times 10^{-5}$. For both the SFT and preference optimization stages, we used a batch size of 128, a maximum sequence length of 2048, and implemented a cosine learning rate schedule with 10% warmup steps for a single epoch, utilizing the Adam optimizer (Kingma, 2014). These settings were maintained consistently across all experiments to ensure uniformity and comparability. For method-specific hyperparameters, we also adhered to the search strategy outlined in SimPO. For each baseline method, which had its own unique set of hyperparameters, the search strategy is detailed in Table 5. Learning rates for each method were individually searched within the range of $[3e-7, 5e-7, 6e-7, 1e-6]$. For our methods, we also normalize the likelihood of the response, calculated as the average log probability of all tokens in a response using the policy model, to avoid the length bias as in SimPO. All training experiments

---

[4]https://huggingface.co/spaces/open-llm-leaderboard-old/open_llm_leaderboard
[5]https://huggingface.co/spaces/open-llm-leaderboard/open_llm_leaderboard

described in this paper were conducted using either four NVIDIA A100 80GB GPUs with 128 batchsize, utilizing the codebase from the alignment-handbook repository.[6]

**Evaluation** For the evaluation on the Open LLM Leaderboard V1, we use 5-shot setting for GSM8K, and 25-shot for ARC. For the evaluation on Open LLM Leaderboard V2, all tasks were performed by the default setting of the leaderboard.

---

```
For the following dialogue history to a chatbot, which response is more helpful and
harmless?

Dialogue history:
{dialogue history}

Response 1:
{Response 1}

Response 2:
{Response 2}

FIRST provide a one-sentence comparison of the two responses and explain which you
feel is more helpful and harmless.  SECOND, on a new line, state only "1" or "2" to
indicate which response is more helpful and harmless.  Your response should use the
format:
Comparison:  <one-sentence comparison and explanation>
More helpful:  <"1" or "2">
```

---

Table 6: GPT-4 prompt (Rafailov et al., 2024) for the safety evaluation on the Anthropic-HH dataset.

## C  FUTURE WORK

This paper presents several exciting directions for future work. First, we aim to develop a deeper theoretical understanding of which density-ratio estimation techniques are most effective for alignment. Our current analysis is limited to the offline setting and does not account for on-policy learning, where the policy interacts with the reward model during training. Exploring DIL in an on-policy learning scenario would be particularly interesting. Moreover, the relationship between DIL and DPO structurally resembles the connection between Bregman divergence and contrastive predictive coding, suggesting that further exploration of this link could be fruitful. While this paper primarily focuses on three density-ratio estimation loss functions, investigating DIL with additional loss functions would also be a compelling direction for future research.

---

[6]https://github.com/huggingface/alignment-handbook

