# OpenReview forum: "On a Connection Between Imitation Learning and RLHF"
_ICLR.cc/2025/Conference — ICLR 2025 Poster_

### Official Review · Reviewer_hdJf · 2024-10-28

**Soundness:** 2
**Presentation:** 2
**Contribution:** 2
**Rating:** 6
**Confidence:** 4

**Summary:**

This paper generalizes preference learning or RLHF frameworks to an imitation learning framework, (DIL). Using this framework they propose multiple offline preference learning methods with different preference modeling such as Bradley-Terry for DPO and LSIF for the best DIL model. Moreover, its performance on benchmarks like Alpaca Eval 2 and Open LLM leaderboard is considerably better than other offline preference learning objectives.

**Strengths:**

* The paper is well-written and easy to follow.

* The first paper to connect RLHF with imitation learning if not mistaken

* Very strong results against popular DPO-variants

**Weaknesses:**

## Is RLHF a form of imitation learning?
Paper frames reward learning as imitation learning and RL as knowledge distillation (KD) and I dont think either of them is correct.

RL: Equation 13 is the reverse KL between the behavior and optimal policy however knowledge distillation is forward KL between teacher (optimal) and student(behavior). KD (forward KL) is distribution fitting or mean seeking whereas reverse KL  is mode seeking which makes the policy focus on high-reward regions rather than fitting the entire distribution with forward KL as in SFT. Overall, the KD claim by the paper is incorrect. Lastly, equation 13 is a known result from DPO paper which is the penultimate step of the optimum solution of equation 14th.


Reward Learning:

In standard RLHF, the reward model is a separate LLM with an additional MLP to predict the scalar reward. So by training a reward model, one does not imitate the expert or optimal policy. What we are doing is fitting a reward model to a predetermined preference model however the caveat is that the optima policy trained by the RL can be parametrized by the reward model trained with which was already proven by the DPO. Lastly, DPO parametrizes the reward model in terms of the policy so when the reward learning objective is trained, we obtain the actual policy.

On the other hand, this paper defines a Boltzmann distribution $\pi_\phi$ (equation 9) in an EBM framework which is the optimal policy induced by the $r_{\phi}(x,y)$. This distribution is maximized on the chosen preferences generated by the $\pi_{expert}$ or imitates $\pi_{expert}$. Following derivations leads to reward likelihood training objectives whereas I am unsure whether $\pi_{ref}$ approximation is free because it introduces rejected responses while IL objective only minimizes on chosen preferences. Nonetheless, this derivation is possible because the $pi_\phi$ has a reward equivalence whereas it does not tell anything for other forms of policy. Overall, I would interpret it as imitating reward rather than policy, not vice versa.

## Direct Imitation Learning
I dont think DIL is novel because it is the backtracking of the derivation of the DPO objective. After all, the 16th equation is the same as the 14th equation of the DPO without the partition and assuming $\pi_{expert} = \pi^*$. DIL is redefining the reward function of DPO, excluding the density ratio estimation part. All in all, I believe this part(excluding density ratio) is already present in DPO.

**Questions:**

Q1) You mention that DIL does not depend on Bradley-Terry but you introduce new reward training with different objectives such as LSIF, UKL, and BCE which are essentially replacements for BT, so doesn't the DIL still rely on some preference modeling assumption?

Q2) In 6.3 you discuss learning dynamics DPO, SimPO, and DIL however Figure 3 does not have DPO, is the discussion from some other paper?

Q3) Do you have additional results on MT-Bench or Arena Hard?

---

> ### Author Response · Authors · 2024-11-23
> **Response to Reviewer hdJf- Part 1**
>
> Dear reviewer hdJf, we appreciate your perception of our contributions on connecting RLHF with imitation learning. We believe that there are some important misunderstandings. Please see our clarifications below:
>
> ---
>
> **Q1. Equation 13 is the reverse KL between the behavior and optimal policy however knowledge distillation is forward KL between teacher (optimal) and student(behavior). KD (forward KL) is distribution fitting or mean seeking whereas reverse KL is mode seeking which makes the policy focus on high-reward regions rather than fitting the entire distribution with forward KL as in SFT. Overall, the KD claim by the paper is incorrect....**
>
>
> **A1.** We apologize for any confusion and believe there may have been some misunderstandings. **The knowledge distillation in Equation (13) that we specifically mentioned here is indeed a "reverse" version, focusing on minimizing the reverse KL divergence, as also mentioned in recent work MiniLLM [1].**
>
> **Yes, Equation (13) is a known result from DPO, but our key contribution is not Equation (13). Please see our response to Q3 for detailed justification why our DIL differs from DPO.**
>
> ---
>
> **Q2. In standard RLHF, the reward model is a separate LLM with an additional MLP to predict the scalar reward. So by training a reward model, one does not imitate the expert or optimal policy.**
>
> **A2.** Thank you for your comments. Although the choice of the loss function and optimization procedure differs, the central aim of our work is to emphasize that the optimal policies of RLHF and DPO are theoretically the same as the imitation learning process. In essence, all these methods aim to discover identical optimal policies, i.e., the chosen response distribution. Specifically,
>
> - Equation (12) shows that imitation learning loss between the distribution chosen response and  energy-based policy is exactly the same as the reward loss based on BT assumption.
>
> - Equation (13) (also shown in DPO)  shows that imitation learning between optimizing policy response and energy-based policy is exactly the same as the RL loss.
>
> - Thus, RLHF with two steps (reward learning and policy learning) can be viewed as conducting  imitation learning between optimizing policy and the distribution of chosen response.
>
> ---
>
> **Q3. I dont think DIL is novel because it is the backtracking of the derivation of the DPO objective. After all, the 16th equation is the same as the 14th equation of the DPO without the partition. DIL is redefining the reward function of DPO, excluding the density ratio estimation part. All in all, I believe this part(excluding density ratio) is already present in DPO.**
>
> **A3.** Thank you for your comments. We believe there are indeed some misunderstandings.  Our work indeed significantly  differs from DPO in several important ways:
>
> - **Theoretical Insight:** We are the first to show that the objective of current alignment methods, such as DPO and RLHF, can theoretically be viewed as fitting the chosen response distribution by minimizing the reverse KL divergence.
> - **General Framework:** We provide a general framework based on Bregman divergence to directly optimize the reverse KL divergence between the policy and the chosen response distribution.
> - **Empirical Results:** We demonstrate that our framework effectively alleviates the decrease in the likelihood of chosen responses and achieves better performance on reasoning-intensive tasks, addressing important limitations in DPO.
>
> ---
>
> **Q4. You mention that DIL does not depend on Bradley-Terry but you introduce new reward training with different objectives such as LSIF, UKL, and BCE which are essentially replacements for BT, so doesn't the DIL still rely on some preference modeling assumption?**
>
> **A4.** Thank you for your comments. As shown in our paper, DIL and DPO indeed share the same assumption, as they both solve the same imitation learning objective (i.e., minimizing the reverse KL divergence between the policy and the chosen response distribution).
>
> However, as demonstrated in our paper, DPO relies on the BT assumption (pairwise loss) to cancel out the normalization term. Due to the self-normalized property in Equation (22), our DIL generalizes to a broad class of objectives that do not rely on pairwise comparisons, unlike DPO/SimPO. Since DPO/SimPO only learn to preserve the relative ordering between the likelihoods of the chosen and rejected responses, they reduce the likelihood of the chosen response, resulting in poor performance in reasoning tasks.

---

> ### Author Response · Authors · 2024-11-23
> **Response to Reviewer hdJf- part 2**
>
> **Q5.  In 6.3 you discuss learning dynamics DPO, SimPO, and DIL however Figure 3 does not have DPO, is the discussion from some other paper?**
>
> **A5.** Thank you for your questions. Yes, the issue of decreasing likelihood of chosen responses in DPO has been widely noticed by many recent works [2,3,4]. Given the superior performance of SimPO over DPO, we only included SimPO in Figure 3 as the motivating example.
>
> ---
>
> **Q6. Do you have additional results on MT-Bench or Arena Hard?**
>
> **A6.** Thank you for your questions.  To answer your question, we have included additional comparisons on Arena-Hard given it is an updated version of MT-Bench in Table 2 (Comparison results on Mistral-7B-Base are also shown in the table below.) We report the win rate (WR) against the baseline model following SimPO. We find our DIL still achieves superior performance on Arena-Hard compared to baselines
>
>
> | Models  | SFT  | DPO  | SLiC | f-DPO | IPO |  CPO| SimPO | DIL |
> | --- | --- | --- | --- | --- | --- | --- | --- | --- |
> | Mistral-7B-Base| 1.3 | 10.4 | 7.3  | 8.1 | 7.5 |5.8 |16.6 | **18.3** |
> | LLama-8B-Base| 3.3 | 15.9 | 10.3  | 14.2 | 17.8 |11.6 |23.4 | **25.6** |
>
>
> **We wholeheartedly appreciate your suggestions and comments. Nevertheless, we believe there are indeed  misunderstandings and not fatal to the major contributions of our manuscript. We have made extensive efforts to address your comments and believe that they adequately address all your concerns.  We believe that the reviewer's insightful comments can be easily and effectively addressed in the final version. Could you please consider increasing your score to reflect the efforts of your review and our rebuttal.**
>
>
> ---
>
>
> [1] MiniLLM: Knowledge Distillation of Large Language Models. ICLR 2024.
>
> [2] Iterative Reasoning Preference Optimization. NeurIPS 2024
>
> [3] Cal-DPO: Calibrated Direct Preference Optimization for Language Model Alignment. NeurIPS 2024.
>
> [4] Smaug: Fixing Failure Modes of Preference Optimisation with DPO-Positive. In Arxiv.

---

> ### Comment · Reviewer_hdJf · 2024-11-24
>
> I have adjusted my score given the reviews and the replies. One last detail is the KTO results are missing despite being mentioned as a baseline in the paper, so it would be better if added.

---

> ### Author Response · Authors · 2024-11-24
> **Further Response to Reviewer hdJf**
>
> Dear Reviewer hdJf,
>
> Thank you very much for carefully reading our response and increasing your score! We are very happy that our response has addressed your comments. We genuinely appreciate your support and will include the new results in the main paper.
>
> Regarding your observation, this was indeed a typo. We mistakenly wrote SLiC as KTO. Nevertheless, we are actively working on comparing our method with KTO on benchmarks.
>
> **We have updated the KTO results in the main Table 2 of the updated submission PDF. The results show that our DIL still achieves better performance than KTO on most widely used benchmarks, particularly on AlpacaEval 2.0 and MATH, demonstrating that DIL can more effectively preserve reasoning abilities compared to KTO while aligning with human preferences.**
>
>
> Sincerely,
>
> The Authors

---

### Official Review · Reviewer_MhY4 · 2024-11-01

**Soundness:** 4
**Presentation:** 3
**Contribution:** 4
**Rating:** 8
**Confidence:** 4

**Summary:**

This paper introduces a new method called Direct Imitation Learning (DIL), which is derived based on an imitation learning perspective on the alignment problem. Specifically, instead of minimizing the forward KL divergence as in SFT, DIL aims to minimize the reverse KL instead. This turns out to require estimating the density ratio $\frac{\pi_{\mathrm{chosen}}}{\pi_{\mathrm{ref}}}$, which the authors show can be done through a Bregman divergence objective. Then, through a similar change-of-variables trick as used in DPO, the authors show that this reward objective can be instead minimized directly in terms of the relevant policies. Hence, the final objective directly optimizes $\pi_{\theta}$ through the Bregman divergence objective.

The authors also show that PPO and DPO can be seen as special cases of the proposed imitation learning formulation. Specifically, reward learning in RLHF can be formulated as a forward KL between $\pi_{\mathrm{chosen}}$ and $\pi_{\phi}$, and the RL step can be seen as a knowledge distillation process (through minimizing a reverse KL) into a final policy $\pi_{\theta}$.

From the experiments side, the authors use the UltraFeedback Binarized dataset for evaluation on the Open LLM Leaderboard and show DIL is generally the best method across the board. For dialogue generation and summarization they use the Anthropic HH dataset and the Reddit TL;DR dataset and show through win rates (as judged by GPT-4) that DIL generally performs best against the SFT, Chosen, and Average responses. Finally, the authors also investigate the likelihood patterns of DIL and SimPO, which generally seem to show that the likelihood of chosen responses stay roughly the same while the likelihood of rejected responses goes down. This is unlike SimPO for which the likelihood of chosen responses also decreases.

**Strengths:**

The paper has several strengths:
1. The paper provides new mathematical connections between imitation learning formulations (various forms of forward and reverse KL optimizations) and previously established RLHF methods like PPO and DPO. As far as I'm aware, these connections are novel and have not been highlighted in past work, making them valuable insights for the community to further build on.
2. To optimize the proposed imitation learning objective, the paper integrates ideas from density ratio estimation [1] and a change-of-variables approach [2] (rewards -> policies) to directly learn the target policy $\pi_{\theta}$, avoiding complexities such as adversarial training.
3. Strong empirical results: the new method DIL seems to generally outperform all baselines in both the Open LLM Leaderboard as well in the summarization and dialogue generation settings.

[1] Masashi Sugiyama, Taiji Suzuki, and Takafumi Kanamori. Density-ratio matching under the bregman divergence: a unified framework of density-ratio estimation. Annals of the Institute of Statistical Mathematics, 64:1009–1044, 2012.

[2] Rafael Rafailov, Archit Sharma, Eric Mitchell, Christopher D Manning, Stefano Ermon, and Chelsea Finn. Direct preference optimization: Your language model is secretly a reward model. Advances in Neural Information Processing Systems, 36, 2024.

**Weaknesses:**

The paper has several weaknesses:
1. While the empirical results seem to consistently outperform prior methods, I’m a bit worried about the statistical significance since the margins seem rather small sometimes (e.g. for Table 2, the improvements are almost always smaller than 1 percentage point). Could the authors include some significance tests or at least standard errors / CIs to provide a better sense of the significance of these improvements?
2. The exposition of the math/theory in the paper could have been a bit clearer (section 4). It took me some time to understand what actually is the final objective that DIL optimizes, and how it came to be. This is because, for example, at the end of section 4.3 the authors state “With the estimated density ratio reward, the surrogate imitation learning objective in Equation (17) can then be solved with any RL algorithms.”, which initially made it seem like DIL would have to resort to RL optimization anyways. But then reading section 4.4 it turns out that’s not what happens and there is actually a different objective that’s maximized (eq. 24). Maybe one thing that could help here is to add a summary box either at the beginning or end of section 4 that summarizes the key steps to go from the general DIL objective (eq. 16) to the formulation in eq. 24.
3. Some parts of the paper require further clarification - please see the Questions section for this.

**Questions:**

1. In Section 5 under the models paragraph, the authors state “For fine-tuning on UltraFeedback Binarized dataset, we use Zephyr-7b-SFT (Tunstall et al., 2023) and Llama3-8b-SFT used in (Meng et al., 2024) as our base models.”, but then in Table 2 the top results are labeled as Mistral-7B-Base. Should that be Zephyr-7B-SFT instead?
2. In Section 5, the authors mention KTO as part of the baselines, but it doesn’t seem the result tables include it? Also, SLiC is included in the result tables, but is not discussed in the baselines paragraph?
3. Could the authors include the base model (SFT) performances in Table 2?
4. In Table 3, what is the difference between Chosen and Average?
5. In Table 3, it might be interesting to compare win rates of DIL directly with DPO or other baselines. Is there a reason the authors didn't include this?
6. At the end of section 6.1, the authors state that “We hypothesize that these improvements can be attributed to avoiding the BT assumption and preventing the decrease in the likelihood of chosen responses.” Could the authors elaborate on why avoiding the BT assumption could lead to these improvements? Do they have examples in mind where BT might not be the right model?
7. I’m a bit confused as to how $\pi_{\mathrm{chosen}}$ is defined. Is it essentially defined to be the policy that, given a preference dataset of $ (x, y_w, y_l) $ triplets, was responsible for generating all the $y_w$ pairs?
8. In the beginning of section 4.3, the authors state that “In the tabular setting, we can directly compute $\pi_{\mathrm{ref}}(y | x)$ and $\pi_{\mathrm{chosen}}(y | x)$.” Could the authors please elaborate on this a bit? It’s not clear to me what the tabular setting here means.
9. Is the Y-axis in figures 1 & 3 the *negative* log likelihood? And for the margins figure on the right, is it a difference of negative log likelihoods? This could use some better labeling. Putting the model name on the y-axis is a bit confusing, and might be better put in the caption.
10. At the end of section 4.1: “achieving this in practice requires full data coverage and infinite models that are rarely met”. What is meant by “infinite models” here?
11. In the paragraph right after equation 22, what’s $\pi_{\mathrm{data}}$?
12. In the paragraph right after equation 22, why is there no log before the reward $r$ in $Z(x)$? Shouldn’t there be since there is one in equation 22 as well?
13. In the paragraph after equation 22, the authors state “This characteristic, determined by the reward definition in Equation (17), is super beneficial as it allows our imitation learning to theoretically generalize to a broader class of loss functions beyond the pairwise BT preference model used in DPO.”. Could the authors please elaborate on this? What does "this characteristic" refer to? And how does it allow the imitation learning to generalize to a broader class of loss functions beyond BT?
14. At the end of section 4.5 the authors state “Specifically, we demonstrate that DPO also falls under the imitation learning objective in Equation (16) and essentially employs the CPC method for density ratio reward estimation.”. While I agree CPC indeed estimates the correct density ratio, it’s unclear to me that this is used in equation 27. Specifically, the learned $f^*$ from equation 26 doesn’t seem to show up in equation 27?
15. Towards the end of 6.1, the authors state “Notably, we observe DPO and SimPO hurt the overall performance in most reasoning-heavy tasks such as GSM8K”. Is this compared to some base model performance? And if so, where is this reported?
16. This statement in 6.1 could use some clarification: “For instance, on LLama3, the improvements are notable on the Math and AlpacaEval 2 benchmarks, with relative gains exceeding 7.5% and 18.2%, respectively.” Is this for DPO or SimPO?

**Details Of Ethics Concerns:**

No concerns.

---

> ### Author Response · Authors · 2024-11-23
> **Response to Reviewer MhY4**
>
> Dear reviewer MhY4, we appreciate your efforts and detailed comments very much! However, we believe that there are some misunderstandings. Therefore, we would like to provide a point-by-point response to your comments.
>
> **Q1. While the empirical results seem to consistently outperform prior methods, I’m a bit worried about the statistical significance since the margins seem rather small sometimes (e.g. for Table 2, the improvements are almost always smaller than 1 percentage point).**
>
>
> **A1.** Thank you for your comments. Please refer to our responses Q6 to Reviewer JHUL, where we provide additional results demonstrating that our improvements are significant compared to SFT, DPO, and SimPO.
>
> ---
>
> **Q2. The exposition of the math/theory in the paper could have been a bit clearer (section 4). It took me some time to understand what actually is the final objective that DIL optimizes, and how it came to be. This is because, for example, at the end of section 4.3 the authors state “With the estimated density ratio reward, the surrogate imitation learning objective in Equation (17) can then be solved with any RL algorithms.”, which initially made it seem like DIL would have to resort to RL optimization anyways…**
>
> **A2.** We apologize for any confusion. What we intend to convey is: "With the estimated density ratio reward, the surrogate imitation learning objective in Equation (17) can be solved using any RL algorithm. However, this two-step process is complex and often unstable."  To address this, in section 4.4, we introduce a simpler approach that directly optimizes the imitation learning objective. This method bypasses the need for RL training and density ratio estimation by leveraging a change-of-variables approach. We have updated the main paper to clearly articulate this.
>
> ---
>
>
> **Response and Clarification to Questions:**
>
> **Thank you for your detailed questions! Below are our responses and clarifications. We will update the submission to include these clarifications.**
>
> 1. Mistral-7B-Base is the same as Zephyr-7B-SFT. We have corrected the typo.
> 2. This was a typo. We mistakenly wrote SLiC as KTO. We have corrected it.
> 3. Yes, we have included the base model (SFT) performance in Table 2 of the updated submission.
> 4. The "average" refers to the average win rates computed by GPT-4 when comparing the SFT-generated responses with the chosen responses in the dataset.
> 5. The reason we chose "vs. SFT" and "vs. Chosen" is to strictly align all settings with those in the original DPO paper.
> 6. As shown in the original DPO objective under the BT assumption, it maximizes the expected relative difference between the implicit 7. rewards of the chosen and rejected responses. Thus, while these methods preserve the relative ordering between the likelihoods of the chosen and rejected responses, they may reduce the absolute likelihood of the chosen response.
> 7. The reviewer is correct. $\pi_{\text{chosen}}$ refers to the probability of generating the chosen response in the preference dataset.
> 8. The tabular setting refers to cases where the state and action spaces are small, and both state functions and action-state functions are represented as tables.
> 9. Yes, it is negative log-likelihood. We apologize for the confusion and have updated the figure accordingly.
> 10. The term “infinite models” was a mistake; it should be “infinite computation.” We apologize for the typo.
> 11. It should indeed be $\pi_{\text{chosen}}$. We apologize for the confusion.
> 12. There should be a log operation here. We apologize for the typo.
> 13. The characteristic refers to the self-normalized property. DPO relies on the BT assumption (pairwise loss) to cancel out the normalization term. Due to this self-normalized property, our imitation learning approach generalizes to a broad class of objectives that do not rely on pairwise comparisons.
> 14. Sorry for your confusion. We also utilize a change-of-variables approach (critic function $f$ to policy $\pi$ using Eqs. (23) and (26)).
> 15. Yes, the results are compared to some base model performance. Please see the updated Table 2, where we provide the performance of the base models (SFT).
> 16. The improvements are over SimPO. We apologize for the confusion and have clarified this in the updated submission.

---

> ### Comment · Reviewer_MhY4 · 2024-11-23
> **Further review**
>
> Thank you for the rebuttal!
>
> > Thank you for your comments. Please refer to our responses Q6 to Reviewer JHUL, where we provide additional results demonstrating that our improvements are significant compared to SFT, DPO, and SimPO.
>
> Thank you for providing these! These do increase my confidence in the signifcance of the results. Nevertheless, could the authors also provide standard errors or CIs?
>
> Also, looking further at Table 2, is it possible SFT should be bolded instead of DIL for Mistral MMLU-PRO? (27.58 > 27.44)
>
> > We apologize for any confusion. What we intend to convey is: "With the estimated density ratio reward, the surrogate imitation learning objective in Equation (17) can be solved using any RL algorithm. However, this two-step process is complex and often unstable." ... We have updated the main paper to clearly articulate this.
>
> Thank you, after having another look, this clarification does help with the reading of the paper!
>
> > Yes, we have included the base model (SFT) performance in Table 2 of the updated submission.
>
> Great, thanks for adding that into the table - it gives a good starting point to compare to for all methods.
>
> > The reason we chose "vs. SFT" and "vs. Chosen" is to strictly align all settings with those in the original DPO paper.
>
> I agree it's generally a good idea to be consistent with prior work. However, that shouldn't restrict the authors from adding additional results here. It makes sense that in the DPO paper, the authors didn't include "vs. DPO", because they'd be comparing against themselves which is maybe not very meaningful. However, in this paper, it makes a lot of sense to do a head-to-head comparison with DPO and maybe SimPO in Table 3, instead of having to infer the strength of one method over the other *indirectly* through the performance against some other baseline (SFT and Chosen in this case).
>
> > Yes, it is negative log-likelihood. We apologize for the confusion and have updated the figure accordingly.
>
> Actually, since it's negative, is it just log-likelihood? The version I'm seeing still seems to just say "likelihood". Though thank you for making the other changes in the figure!
>
> > Sorry for your confusion. We also utilize a change-of-variables approach (critic function  to policy  using Eqs. (23) and (26)).
>
> I'm still a bit confused here. Are the authors plugging the result from Eq. 26 ($f^*(x, y) / \beta = \log \frac{\pi_{\mathrm{chosen}}(y | x)}{\pi_{\mathrm{ref}}(y | x) c(x)}$) into Eq. 25? If so, then you don't even need Eq. 23, since Eq. 26 gives essentially the same result? If the authors could clarify their exact steps in a bit more detail, that would be helpful.
>
> ### Further questions
> 1. Why is DPO not included in Fig. 1 & 3? From the reply to some of the other reviewers I understand it's because SimPO is the motivating example, but I still think adding DPO could help put the results in those figures in better perspective, especially since DPO is still widely used.
>
> 2. In 4.1, why is it justified to take $\pi_{\mathrm{ref}}(y | x) = 0.5 \mathbb{I}(Y = y_l) + 0.5 \mathbb{I}(Y = y_w)$ for sampling from the reference distribution?
>
>
> Finally, thanks for taking the time to address all the other minor clarifications/typos!

---

> > ### Author Response · Authors · 2024-11-24
> > **Further Response to Reviewer MhY4**
> >
> > **Thank you for your additional comments to facilitate further discussion, which will further improve our paper.  Please find our responses below.**
> >
> > **Q1. Thank you for providing these! These do increase my confidence in the significance of the results. Nevertheless, could the authors also provide standard errors or CIs?**
> >
> > **A1.** Yes. Thank you for your suggestion!  In the following table, we provided the performance with standard deviation of our DIL and state-of-the-art baselines on Mistral-7B-Base. We can observe that our DIL does significantly outperform the baselines across multiple benchmarks. Specifically, DIL achieves the highest performance on HumanEval, BBH, MUSR, MATH, GSM8K, and AlpacaEval 2, with lower standard deviations in most cases, indicating both superior and more stable performance compared to DPO and SimPO. These results demonstrate the robustness and effectiveness of our approach, further validating our contributions to preference alignment.
> >
> > | Methods  | HumanEval | &nbsp; BBH  | MUSR  | MATH | GSM8K | AlpacaEval 2 |
> > | --- | --- | --- | --- | --- |--- |--- |
> > | DPO   | 31.7 $\pm$ 0.3 | 43.27  $\pm$ 0.4 | 43.65 $\pm$ 0.3 | 1.36  $\pm$ 0.4 | 21.76  $\pm$ 1.2 | 12.5 $\pm$ 0.2 |
> > | SimPO   | 26.5  $\pm$ 0.6 | 42.94  $\pm$ 0.3 | 39.68 $\pm$ 0.5 | 2.49  $\pm$ 0.5 | 22.21  $\pm$ 1.5 | 20.8 $\pm$ 0.1 |
> > | DIL   | **33.5 $\pm$ 0.5** | **43.59 $\pm$ 0.5** | **44.05 $\pm$ 0.3** | **2.95 $\pm$ 0.3** | **32.19 $\pm$ 1.1** | **21.7 $\pm$ 0.2** |
> > ---
> >
> > **Q2. Is it possible SFT should be bolded instead of DIL for Mistral MMLU-PRO? (27.58 > 27.44)**
> >
> > **A2.** Thank you for pointing out the typo! We have corrected it.
> >
> > ---
> >
> > **Q3. However, in this paper, it makes a lot of sense to do a head-to-head comparison with DPO and maybe SimPO in Table 3, instead of having to infer the strength of one method over the other indirectly through the performance against some other baseline (SFT and Chosen in this case).**
> >
> > **A3.** Thank you for your insightful comments. To address it, we provide the head-to-head win rates comparison with DPO and SimPO on both TL;DR summarization and Anthropic-HH. We can observe that our proposed DIL achieves higher head-to-head win rates compared to both DPO and SimPO on these tasks, as shown in the following table. Thank you again for your valuable feedback, and we hope these additional results address your comments thoroughly.
> >
> > |  Datasets | TL;DR summarization  | Anthropic-HH  |
> > | --- | --- | --- |
> > | DIL v.s. DPO    |  59.4% |  62.8% |
> > | DIL v.s. SimPO |  58.7% |  63.3% |
> >
> > ---
> >
> > **Q4. Actually, since it's negative, is it just log-likelihood?**
> >
> > **A4.** Sorry for the confusion. Yes. It is log-likelihood, not negative log-likelihood. We apologize for any misunderstanding.
> >
> > ---
> >
> > **Q5. If the authors could clarify their exact steps in a bit more detail, that would be helpful.**
> >
> > **A5.** Sorry for the confusion. We did indeed make a typo in Equation (26). The $r$ should be $r^*$. Using $r^*$ in both Equation (26) and Equation (23) establishes a connection between the policy $\pi^*$ and the critic function $f^*$. Thus, we can use the policy $\pi$ to represent the critic function $f$ in Equation (25) by leveraging a change-of-variables approach, resulting in the final loss function in Equation (27).
> >
> > ---
> >
> >
> > **Response to Further Questions**
> >
> > 1. We completely agree with the reviewer that including DPO would be beneficial. Following your suggestions, we have updated the DPO training dynamics results on Mistral-7B-Base in Figure 3 of the updated submission. (Given the short rebuttal period and limited computational resources, we are committed to including the results on LLaMA-8B in the final version.) We observe that the likelihood of both chosen and rejected responses continues to decrease in DPO, which aligns with observations from many recent works [1, 2, 3].
> > 2. There are a number of choices for sampling from the reference policy in EBMs, but this particular choice simplifies the EBMs and has been found to produce stable results in practice, as demonstrated in recent works [4, 5].
> >
> > **Dear Reviewer MhY4, We gratefully appreciate your time in reviewing our paper and your comments. We have made extensive efforts to address your comments. If the reviewer's concerns are clarified, we would be grateful if the reviewer could increase the score. Many thanks for your time!**
> >
> > [1] Iterative Reasoning Preference Optimization. NeurIPS 2024
> >
> > [2] Cal-DPO: Calibrated Direct Preference Optimization for Language Model Alignment. NeurIPS 2024.
> >
> > [3] Smaug: Fixing Failure Modes of Preference Optimisation with DPO-Positive. In Arxiv.
> >
> > [4] Concept Learning with Energy-Based Models. In Arxiv.
> >
> > [5] Self-Play with Adversarial Critic: Provable and Scalable Offline Alignment for Language Models. In Arxiv.

---

> > > ### Comment · Reviewer_MhY4 · 2024-11-24
> > > **Thank you**
> > >
> > > Thank you for the detailed response! Most of my concerns have been addressed, and hence I've increased the score.
> > >
> > > It would be great if the authors could include both of the above tables (standard deviations & head-to-head with DPO/SimPO) in a final version of their paper.

---

> > > > ### Author Response · Authors · 2024-11-24
> > > > **Thank you very much for your reply**
> > > >
> > > > Dear Reviewer MhY4,
> > > >
> > > > We are very glad to hear that our rebuttal and the discussion have adequately addressed your concerns.
> > > >
> > > > Thank you as well for your additional comments, which have facilitated further discussion and will help improve our paper.
> > > >
> > > > We will ensure that our discussion and the additional results are included in the final version of our paper.
> > > >
> > > > Best regards,
> > > >
> > > > The Authors

---

### Official Review · Reviewer_sbFs · 2024-11-04

**Soundness:** 3
**Presentation:** 4
**Contribution:** 4
**Rating:** 8
**Confidence:** 4

**Summary:**

- This paper reinterprets preference alignments methods like RLHF and DPO as special cases of a more general imitation learning objective.
 - They mathematically show how the RLHF and DPO objective functions fit within a general imitation learning framework.
 - They develop a new alignment method DIL based on imitation learning with the objective as minimizing the reverse KL loss between the optimal policy and current policy and derive a preference data based learning objective which suppresses the likelihood of generating dispreferred responses while increasing the likelihood of generating preferred responses.
 - They empirically show that DIL results in a better policy compared to other offline alignment methods across reasoning and alignment benchmarks.

**Strengths:**

- This paper is well written and presents intriguing connections between imitation learning and human preference alignment.
 - They derive a new alignment framework based on imitation learning and show empirical improvements on existing baseline.
 - DIL shows significantly better training dynamics compared to SimPO by ensuring that the likelihood of generating chosen responses is maintained.

**Weaknesses:**

- The amount of data needed for satisfactory alignment with DIL compared to other methods is not clear. The authors claim that DIL is more efficient, so it would be nice to see some metrics that measure this.
 - All the models in the experiments are smaller (<10B parameters) so it’s not clear how effective DIL would be for larger models.

**Questions:**

- Since DIL doesn’t suppress the likelihood of dispreferred responses as much as SimPO, how does this affect alignment from a safety perspective? Is the model more prone to generate harmful responses?

---

> ### Author Response · Authors · 2024-11-23
> **Response to Reviewer sbFs**
>
> Dear reviewer sbFs, we appreciate the reviewer's perception of our contributions to both empirical and theoretical analysis, and we thank the reviewer for their insightful questions. Please find our detailed responses below:
>
>
> **Q1. The amount of data needed for satisfactory alignment with DIL compared to other methods is not clear.**
>
> **A1.** Thank you for your comments. We apologize for any confusion. The term "efficient" does not refer to data efficiency; instead, it highlights that, compared to PPO in RLHF, our DIL approach is more efficient in terms of computation, speed, and engineering effort. DIL eliminates the need for an additional stage of training a reward model and, during policy training, does not require decoding online responses (which is typically slow) or training an additional value model.
>
> ---
>
> **Q2. All the models in the experiments are smaller (<10B parameters) so it’s not clear how effective DIL would be for larger models.**
>
> **A2.** Thank you for your suggestions. Following your suggestion, we conducted additional experiments on Mixtral-8x22B-Instruct-v0.1. From the following results, we find that the DIL still achieves better performance compared to baselines. These results have also been included in Table 8 in the updated submission:
>
> | Mixtral-8x22B  | HumanEval  | LeetCode  | MATH | TheoremQA |
> | --- | --- | --- | --- | --- |
> | DPO | 75.1 | 24.5 | 48.5 | 34.7 |
> | SimPO | 76.2 | 22.5  | 50.3 | 35.5 |
> | DIL | **77.3** | **28.7**  | **52.8**  | **36.9** |
>
> ---
> **Q3. Since DIL doesn’t suppress the likelihood of dispreferred responses as much as SimPO, how does this affect alignment from a safety perspective? Is the model more prone to generate harmful responses?**
>
> **A3.** Thank you for your questions. Based on the results on Anthropic Helpful and Harmless, we observe that our DIL achieves significantly better performance than SimPO, demonstrating that DIL is less prone to generating harmful responses. We hypothesize that there may be two potential reasons for this:
> - (i) Although DIL does not suppress the likelihood of dispreferred responses, the margin likelihood between preferred and dispreferred responses in DIL is at the same level as SimPO, demonstrating that DIL still has the capability to distinguish between preferred and dispreferred responses.
> - (ii) Additionally, not all rejected responses in preference datasets are low-quality. It may be the case that both chosen and rejected responses are high-quality, but the chosen response is slightly better. In this case, it may not be ideal for the model to excessively reduce the likelihood of rejected responses.

---

> > ### Comment · Reviewer_sbFs · 2024-11-24
> > **Thank you to the authors**
> >
> > Thank you for answering all my questions!

---

> > > ### Author Response · Authors · 2024-11-25
> > > **Thank you for your response**
> > >
> > > Dear Reviewer sbFs,
> > >
> > > Thank you very much for reviewing our paper and reading our rebuttal. We sincerely appreciate your recognition of our contribution!
> > >
> > > We are truly grateful for your time and your reply.
> > >
> > > Best regards, Authors

---

### Official Review · Reviewer_JHUL · 2024-11-08

**Soundness:** 3
**Presentation:** 2
**Contribution:** 2
**Rating:** 6
**Confidence:** 4

**Summary:**

The paper makes a connection between various approaches for RLHF of large language models and imitation learning. In particular the authors re-derive a well known connection between probabilistic inference and reinforcement learning which associates the reward function with the energy of a Boltzmann distribution (see e.g. [1] for a good review of all the related methods and derivations) for the special case of RLHF.
From this perspective classical reward model learning can be derived as matching the energy to the generated responses with highest reward. Based on this the authors then derive a surrogate objective (DIL) that is closely related to DPO and other RLHF algorithms that exist, but which makes less assumptions on the form of the reward model. They show empirical evaluations on language modeling which match/gives slight improvement over DPO.

[1] Levine, Sergey. "Reinforcement learning and control as probabilistic inference: Tutorial and review." arXiv preprint arXiv:1805.00909 (2018).

**Strengths:**

- The connection between RLHF and imitation learning approaches is highly relevant to the community and the first part of the paper (background and initial derivation up to Eq.12-14) is well presented and leaves the reader with a condensed and improved understanding of how different existing algorithms relate (although perhaps more references to the literature could help, see weaknesses below).
- Any improvement over DPO (which is perhaps the predominant algorithm at least for offline RLHF from fixed datasets) is relevant to the community.
- The benchmarks used are open and relevant and at reasonable scale (i.e. 7B models)

**Weaknesses:**

- My first main gripe with the paper is that the idea that RLHF in it's entirety is performing imitation learning seems to stand on shaky foundations. A lot of leaps from one objective to another are required to arrive at this conclusion and a lot of the nuances in the differences between different objectives get lost along the way (that are already well discussed in the literature see e.g. Sergey Levine's review and also existing literature on offline RL as probabilistic inference). For example, the title says "RLHF secretly performs imitation learning" then up to Eq. 12 this thread is followed closely, and I find the connection that is made between reward model learning and the imitation learning perspective insightful, however directly after the authors make a leap to knowledge distillation / minimizing the reverse KL, which then attains the actual RL objective. This objective then is no longer directly related to learning from a dataset of reference or "chosen" examples (as would be the case in imitation learning) but instead can be understood as imitating an optimal policy (and not any policy that generated the dataset) on the state distribution induced by the currently learned policy (see also [3]). It thus really is RL (and not just imitation learning) and has to "deal" with all the problems RL comes with, i.e. exploration of the energy landscape of the optimal policy is required, premature convergence could be an issue etc.. The fact that the energy itself is given by a reward model that comes from matching chosen examples on a pre-collected dataset has no bearing on this. This is easy to see as depending on the temperature (which also pops out wihtout explanation) chosen in Eq 13. the policy may collapse to matching a single mode of the energy model but may also result in much higher energy / better reward than the chosen policy. The authors do discuss some of these nuances below in a short section on why SFT (which uses a forward KL) might underperform the reverse KL approach. But all this does is it leaves the reader with the impression that the authors painted too broad a picture to derive a connection that then, in practice is not relevant. This could be rectified by perhaps framing the paper as "RLHF can be seen as imitating an optimal policy based on human preferences" and toning down some of the quite strong language, e.g. "learning without an RL loop" etc.
- The paper seems to be derived backwards as some of connections made feel slightly contrived upon reading. E.g. the jump from the original objective to knowledge distillation mentioned above. The steps taken to arrive at a DPO like objective from density ratio estimation etc. The paper requires a lot of steps to arrive at a simple algorithm that the authors probably had in mind from the get-go and started from.
- The knowledge distillation connection seems tenuous (and already known), it seems more straightforward to think of the entire process as imitating a better policy as in MARWIL [2] or as chaining a specific variant of an improvement operator and distillation as already derived in detail for many different variants in [3].
- A lot of the derived formulas and connections are already known in the literature but this is often not explicitly stated, e.g. "
In this section, we connect RLHF to the imitation learning framework. We show that RLHF is a special
case of imitation learning problem by defining the following specialized energy-based mode" in front of Eq 9, which very clearly is already derived in the DPO paper and literature on RL as probabilistic inference. It is fine to re-state such derivations but then please say: we built on a well known connection between RL and energy based models/probabilistic inference.
- The key innovation that the paper hinges on seems to be the approximation of the log ratio between chosen and current policy but the derivation seems very ad-hoc and on shaky foundations. To be explicit: in order to arrive at their Eq. 21 (and thus Eq 24 which is their DIL objective) they make the assumption that the reference policy is the same as the policy that generates the rejected samples only and disregard any terms on the positive examples; i.e. "Here, we use the set of rejected responses y_l ∼ π_ref(y | x) to approximate the expectations under π_ref(y | x)". This is simply a wrong assumption. I do not know why the authors have chosen to make the assumption, but it feels like a contrived way to come to Equation 24 and form a connection to a DPO like objective.
- The results are on relevant benchmarks, but the improvement over DPO seems minor in most cases. In this scenario what would be nice would be to analyze qualitative differences, e.g. examples which DIL seems to have stronger performances on compared to DPO. Or an analysis on how closeness (in KL) wrt. to the reference policy evolves during the course of optimization for different algorithms and how this affects performance. Or a plot that have the DIL objective on the x axis and win-rate (over different models, e.g. reference policy and DPO) on the Y-axis.

[2] Wang, Qing, et al. "Exponentially weighted imitation learning for batched historical data." Advances in Neural Information Processing Systems 31 (2018).
[3] Ghosh, Dibya, Marlos C Machado, and Nicolas Le Roux. "An operator view of policy gradient methods." Advances in Neural Information Processing Systems 33 (2020): 3397-3406.

**Questions:**

A discussion and answers regarding the weaknesses listed above would be appreciated. And if the authors can provide some more rationale and clean-up the derivations the score could be improved.

---

> ### Author Response · Authors · 2024-11-23
> **Response to Reviewer JHUL- part 1**
>
> Dear reviewer JHUL, we appreciate your efforts and detailed comments very much! However, we believe that there are some misunderstandings. Therefore, we would like to provide a point-by-point response to your comments.
>
> ---
> **Q1. My first main gripe with the paper is that the idea that RLHF in its entirety is performing imitation learning seems to stand on shaky foundations. A lot of leaps from one objective to another are required to arrive at this conclusion and a lot of the nuances in the differences between different objectives get lost along the way …**
>
> **A1.**  Thank you very much for your insightful comments!
>
> We agree that the title "RLHF secretly performs imitation learning" may overstate the case, particularly given the nuances of the transition between imitation learning and reinforcement learning within the RLHF pipeline. **We have revised the title in the updated submission to more accurately reflect this perspective: “DIL: Direct Imitation Learning for Preference Alignment and Connections to RLHF.”**
>
> **Although the choice of the loss function and optimization procedure differs, the central aim of our work is to emphasize that the optimal policies of RLHF and DPO are theoretically the same as the imitation learning process, they aim to discover identical optimal policies, i.e., the chosen response distribution. We would like to kindly remind the reviewer of our contributions compared to works based on RL as inference [1].**
>
> - **Theoretical Insight:** We are the first to show that the objective of current alignment methods, such as DPO and RLHF, can theoretically be viewed as fitting the chosen response distribution by minimizing the reverse KL divergence.
> - **General Framework:** We provide a general framework based on Bregman divergence to directly optimize the reverse KL divergence between the policy and the chosen response distribution.
> - **Empirical Results**: We demonstrate that our framework effectively alleviates the decrease in the likelihood of chosen responses and achieves better performance on reasoning-intensive tasks, addressing important limitations in DPO.
>
> [1]  Reinforcement learning and control as probabilistic inference: Tutorial and review In ArXiv.
>
> ---
>
> **Q2. The paper seems to be derived backwards as some of the connections made feel slightly contrived upon reading. E.g. the jump from the original objective to knowledge distillation mentioned above. The steps taken to arrive at a DPO like objective from density ratio estimation etc.**
>
> **A2.** Thank you for your comments. The knowledge distillation steps we provided are intended to demonstrate that the standard KL-regularized RLHF problem and DPO both aim to discover the same optimal policies as those obtained by conducting imitation learning on the chosen response using reverse KL divergence.
>
> Based on these insights, we propose a more general framework based on Bregman divergence to directly optimize the reverse KL divergence between the policy and the chosen response distribution. We have updated the paper to clarify the derivation step further.
>
> ---
> **Q3. The knowledge distillation connection seems tenuous (and already known), it seems more straightforward to think of the entire process as imitating a better policy as in MARWIL [2] or as chaining a specific variant of an improvement operator and distillation as already derived in detail for many different variants in [3].**
>
> **A3.** Thank you for your comments and mentioning these related works. We totally agree with the reviewer that knowledge distillation is well-known. However,  The knowledge distillation steps we provided are intended to show that the standard KL-regularized RLHF problem and DPO both aim to discover the same optimal policies as conducting imitation learning on the chosen response with reverse KL divergence, which are inherently different from these works as also shown in our response to Q1. We will discuss more past works on knowledge distillation  in the updated submission.
>
>
> **Q4. E.g. " In this section, we connect RLHF to the imitation learning framework. We show that RLHF is a special case of imitation learning problem by defining the following specialized energy-based mode" in front of Eq 9, which very clearly is already derived in the DPO paper and literature on RL as probabilistic inference. It is fine to re-state such derivations but then please say: we built on a well known connection between RL and energy based models/probabilistic inference.**
>
>
> **A4.** Thank you for your suggestions. We totally agree with the reviewer that the energy-based model in Eq. 9 is well-known; however, our contribution lies in establishing the connection between RLHF/DPO and imitation learning over the chosen response in the preference data (see our detailed response to Q1). We have updated the paper and rephrased this sentence as: “We build our analysis on a well-known connection between RL and energy-based models.” We apologize for your confusion.

---

> ### Author Response · Authors · 2024-11-23
> **Response to Reviewer JHUL- part 2**
>
> **Q5. "Here, we use the set of rejected responses $y_l ∼ \pi_{ref}(y | x)$ to approximate the expectations under $\pi_{ref}(y | x)$". This is simply a wrong assumption. I do not know why the authors have chosen to make the assumption, but it feels like a contrived way to come to Equation 24 and form a connection to a DPO like objective.**
>
> **A5.** Thank you for your insightful comments! We believe there may be some misunderstandings. The assumption is indeed reasonable. It is acceptable to use the set of rejected responses to approximate the expectations since both chosen and rejected responses  are indeed sampled from  $\pi_{ref}(y | x)$, as also demonstrated in [1,2].
>
> Furthermore, we can use both chosen and rejected responses to approximate this expectation.
> We acknowledge that this approximation may have bias, and there are indeed many ways to approximate the expectations [1]. However, we have shown that with this approximation, DIL achieves better performance, as demonstrated in the following table. Additionally, we demonstrate that DPO with CPC as a density estimation method can be viewed as an imitation learning objective using this approximation, which we believe provides valuable insights. We have updated the paper to clarify this.
>
> | Methods  | HumanEval  | LeetCode  | GSM8K | MATH | TheoremQA | AlpacaEval2.0 |
> | --- | --- | --- | --- | --- |--- | --- |
> | Chosen+Rejected | 29.5 | 2.9  |23.6 | 2.1 | 9.3 | 15.8 |
> | Rejected | **33.5** | **3.4** | **32.2**  | **3.0** | **12.5** | **21.7** |
>
> From the table, we can observe that utilizing rejected responses to approximate the expectation achieves better performance, which is reasonable as the goal is to decrease the likelihood of the rejected response rather than the chosen one.
>
>
> [1] Self-Play with Adversarial Critic: Provable and Scalable Offline Alignment for Language Models. In Arxiv
>
> [2] Direct Preference Optimization: Your Language Model is Secretly a Reward Model. In NeurIPS 2023
>
> ---
>
> **Q6. The results are on relevant benchmarks, but the improvement over DPO seems minor in most cases.**
>
> **A6.** Thank you for your excellent suggestions regarding potential additional experiments to justify the improvements over DPO. We want to emphasize that the improvements over DPO are indeed significant, especially on reasoning-heavy tasks. Recently, many researchers have observed that DPO generally decreases downstream task performance, particularly on reasoning-heavy tasks like Math and Coding. To verify this, we have included additional comparisons with DPO on more reasoning-heavy tasks, such as Coding (HumanEval, LeetCode, MBPP) and Math (GSM8K, MATH, TheoremQA).
>
> | Methods  | HumanEval  | LeetCode  | GSM8K | MATH | TheoremQA | AlpacaEval2.0 |
> | --- | --- | --- | --- | --- |--- | --- |
> | Mixtral-7B-Base (SFT) | 28.1 | 3.3  | 28.1 | 2.3 | 7.0 | 6.2 |
> | DPO | 31.7 | 2.2  |21.7 | 1.4 | 9.8 | 12.5 |
> | SimPO | 26.5 | 1.9  |22.2 | 2.5 | 8.5 | 20.8 |
> | DIL | **33.5** | **3.4** | **32.2**  | **3.0** | **12.5** | **21.7** |
>
>
> From the table, we observe that DIL achieves significant improvements over DPO. Moreover, it demonstrates that DIL more effectively preserves reasoning abilities, such as the mathematical and abstract reasoning skills of the base SFT model, and even significantly outperforms SFT in many cases. Consequently, DIL imposes a lower alignment tax, given its strong performance on the preference alignment benchmark, AlpacaEval 2.0.
>
> ---
>
> **We gratefully appreciate your time in reviewing our paper and your comments. We have made extensive efforts to address your comments and believe that they adequately address all your comments. The reviewer's comments are mainly about some clarifications and are not fatal to the contributions of our manuscript; we believe that the reviewer's insightful comments can be easily and effectively addressed in the final version. We would be grateful if the reviewer could increase the score.**

---

> > ### Author Response · Authors · 2024-11-24
> > **A sincere and kind reminder to the ICLR Reviewer JHUL**
> >
> > Dear ICLR Reviewer JHUL,
> >
> > We greatly appreciate your time and the insightful comments provided during the review of our paper.
> >
> > We have made extensive efforts to address all your questions, suggestions, and misunderstandings in the response and believe that they adequately address all your concerns. We believe that the reviewer's insightful comments can be easily and effectively addressed in the final version.
> >
> > With the discussion phase ending soon, we would like to confirm whether there are any other clarifications they would like. We would be grateful if the reviewer could increase the score.
> >
> > Thank you again for your time and valuable input; we are deeply appreciative.
> >
> > Best regards,
> >
> > The Authors

---

> > ### Comment · Area_Chair_Ss4z · 2024-11-28
> > **Concerns addressed?**
> >
> > It'd be great if you can respond to the author's rebuttal and convey whether your opinion about this paper has changed.

---

> > > ### Author Response · Authors · 2024-11-30
> > > **Kind reminder to Reviewer JHUL (discussion deadline is approaching)**
> > >
> > > Dear ICLR Reviewer JHUL,
> > >
> > > Thank you again for your time and efforts in reviewing our paper. We have carefully responded to each of your questions.
> > >
> > > Given that the author-reviewer discussion deadline is approaching, we would greatly appreciate it if you could kindly review our responses and share your valuable feedback. We would be grateful if you could consider increasing the score given our clarification.
> > >
> > > Thank you very much!
> > >
> > > Best regards,
> > >
> > > The Authors

---

> ### Author Response · Authors · 2024-11-25
> **Kind reminder to Reviewer JHUL**
>
> Dear ICLR Reviewer JHUL,
>
> We greatly appreciate your time and the insightful comments provided during the review of our paper.
>
> We have made extensive efforts to address all your questions, suggestions, and misunderstandings in our response and believe that they adequately address your concerns. The reviewer's comments primarily focused on clarifying certain claims and experimental details. We have addressed these points in our response by providing detailed explanations, including clarifications on specific claims and additional results. We believe that the reviewer's insightful comments can be effectively and easily  addressed in the final version.
>
> **As you and other reviewers have noted, our work offers valuable insights, builds a very interesting connection between imitation learning and RLHF, and presents a practical effective framework. Since your comments do not impact the major contributions of our manuscript, we would be grateful if you could consider increasing the score.**
>
> We are extremely grateful for your time.
>
> Best regards,
>
> The Authors

---

> ### Author Response · Authors · 2024-11-26
> **Kind reminder to Reviewer JHUL**
>
> Dear ICLR Reviewer JHUL,
>
> We greatly appreciate your time and the insightful comments. We have made extensive efforts to address all your questions, suggestions, and misunderstandings in our response and believe that we have addressed your concerns.
>
> **In your original review, you mentioned, “If the authors can provide some more rationale and clean up the derivations, the score could be improved.”**
>
> Thus, we sincerely want to confirm if there are any additional clarifications you would really like us to address. We would be grateful if you could consider increasing the score.
>
>
> Best regards,
>
> The Authors

---

> ### Comment · Reviewer_JHUL · 2024-12-01
> **Thanks for the responses!**
>
> Thanks to the authors for adjusting the paper and replying to my concerns. Most of the minor concerns have been addressed and the additional results and ablations (e.g. re. the choice of sampling dataset for estimating Eq 24) are appreciated.
>
> I think the paper is improved and a bit clearer, especially putting emphasis more on connections between RLHF and IL seems a good choice in the title.
>
> I still maintain that some of the derivations are lengthy and the fact that the ablations reveal that the choice of the sampling dataset for Eq. 24 makes a huge difference (and is a choice that is clearly made to make the algorithm resemble DPO as much as possible) makes me feel that the entire derivation could have been much shortened and the paper just been presented as a better alternative to DPO.
>
> Nonetheless the paper does provide potentially useful insights to the community and in its revised version could be considered for publication as is (albeit a larger rewrite would probably improve it further by quite a bit). I have thus adjusted my score upwards.

---

> ### Author Response · Authors · 2024-12-01
> **Thank you for your support!**
>
> Dear ICLR Reviewer JHUL,
>
> We appreciate your recognition of our contribution and agree that our current version addresses your concerns and could be considered for publication as is.
>
> Our work provides an intriguing explanation and insight for DPO/RLHF from the perspective of imitation learning and presents a new alignment objective inspired by this insight. We will shorten the entire derivation following your suggestion.
>
> Thank you once again for your thoughtful review and valuable feedback!
>
> Sincerely,
>
> The Authors

---

### Author Response · Authors · 2024-11-21
**Official Comment by Authors**

We would like to thank all the reviewers for their thoughtful feedback and time.

We deeply appreciate the numerous positive comments on our work, such as describing it as "valuable insights," "solid motivations," and "solid theoretical and empirical analysis".

The main comments from the reviewers relate to some misunderstandings, clarifications, and the need for additional minor experiments.

We have made our greatest efforts to prepare a point-by-point response to each reviewer.

Thank you again for your time.

Best regards,

The Authors

---

### Meta-Review · Area_Chair_Ss4z · 2024-12-20

**Metareview:**

This paper presents Direct Imitation Learning (DIL), a novel framework for aligning large language models with human preferences. The authors reinterpret existing alignment methods like RLHF and DPO as special cases of imitation learning and introduce a new objective function based on minimizing the reverse KL divergence between the model's policy and the distribution of chosen responses.

The paper's strengths are its solid theoretical foundation and strong empirical results. The authors demonstrate that DIL outperforms existing methods on several benchmarks, including the Open LLM Leaderboard and AlpacaEval 2.0.  However, the paper has some weaknesses. The reviewers pointed out that the connection between RLHF and imitation learning is not entirely novel, and some derivations could be more concise. Additionally, the reviewers raised concerns about the clarity of the presentation and the significance of the empirical improvements. Most of these weaknesses were addressed during the rebuttal period, making this work a good contribution to ICLR.

**Additional Comments On Reviewer Discussion:**

During the rebuttal period, the authors addressed reviewers' concerns by clarifications, providing additional experimental results, and improving the presentation. They also emphasized the novelty of their work by highlighting the connection between RLHF and imitation learning, which provides a new perspective on alignment research. The reviewers agreed that the paper should be accepted after the authors addressed their concerns during the rebuttal period

---

### Decision · Program_Chairs · 2025-01-22

Accept (Poster)